# Development of a functional salivary gland tissue chip with potential for high-content drug screening

Yuanhui Song[1,2,13], Hitoshi Uchida[2,3,13], Azmeer Sharipol[1,2,13], Lindsay Piraino [1,2,4], Jared A. Mereness[1,2,5], Matthew H. Ingalls[2,3], Jonathan Rebhahn [6], Shawn D. Newlands[7,8,9], Lisa A. DeLouise[1,2,4,10], Catherine E. Ovitt[2,3,8] & Danielle S. W. Benoit [1,2,3,5,8,10,11,12✉]

Radiation therapy for head and neck cancers causes salivary gland dysfunction leading to permanent xerostomia. Limited progress in the discovery of new therapeutic strategies is attributed to the lack of in vitro models that mimic salivary gland function and allow high-throughput drug screening. We address this limitation by combining engineered extracellular matrices with microbubble (MB) array technology to develop functional tissue mimetics for mouse and human salivary glands. We demonstrate that mouse and human salivary tissues encapsulated within matrix metalloproteinase-degradable poly(ethylene glycol) hydrogels formed in MB arrays are viable, express key salivary gland markers, and exhibit polarized localization of functional proteins. The salivary gland mimetics (SGm) respond to calcium signaling agonists and secrete salivary proteins. SGm were then used to evaluate radio-sensitivity and mitigation of radiation damage using a radioprotective compound. Altogether, SGm exhibit phenotypic and functional parameters of salivary glands, and provide an enabling technology for high-content/throughput drug testing.

[1] Department of Biomedical Engineering, University of Rochester, Rochester, NY, USA. [2] Center for Oral Biology, University of Rochester Medical Center, Rochester, NY, USA. [3] Department of Biomedical Genetics, University of Rochester Medical Center, Rochester, NY, USA. [4] Department of Dermatology, University of Rochester Medical Center, Rochester, NY, USA. [5] Department of Environmental Medicine, University of Rochester Medical Center, Rochester, NY, USA. [6] David H. Smith Center for Vaccine Biology and Immunology, University of Rochester Medical Center, Rochester, NY, USA. [7] Department of Otolaryngology, University of Rochester Medical Center, Rochester, NY, USA. [8] Wilmot Cancer Institute, University of Rochester Medical Center, Rochester, NY, USA. [9] Department of Neuroscience, University of Rochester Medical Center, Rochester, NY, USA. [10] Materials Science Program, University of Rochester, Rochester, NY, USA. [11] Department of Chemical Engineering, University of Rochester, Rochester, NY, USA. [12] Center for Musculoskeletal Research, University of Rochester Medical Center, Rochester, NY, USA. [13] These authors contributed equally: Yuanhui Song, Hitoshi Uchida, Azmeer Sharipol. ✉email: benoit@bme.rochester.edu

Saliva is essential for oral health and homeostasis. A decrease in saliva secretion, known as xerostomia, has debilitating consequences on the oral mucosa, dentition, and basic activities, such as eating and speaking. Reduced salivation frequently results from various medications including >500 anti-hypertensives, diuretics, and antihistamines; simultaneous use of multiple medications can increase its severity[1,2]. While drug-induced dry mouth is reversible if the medication(s) is stopped or an alternative pharmacologic exists, a permanent loss of saliva is caused by radiation therapy, which is commonly used to treat nearly 600,000 head and neck cancer patients worldwide annually[3]. Salivary gland function is permanently impaired by radiation due to the irreversible loss of secretory acinar cells[4–7]. Efforts to uncover the basis for salivary gland radiosensitivity and to identify effective radioprotective agents have been limited by the lack of suitable in vitro salivary gland models. Culture of primary salivary gland cells is accompanied by a rapid loss of the secretory acinar cell phenotype[8–10]. Therefore, the major goal of this work was to develop a functional salivary gland tissue mimetic for high-content and high-throughput testing.

The microenvironment is critical for epithelial tissue morphogenesis and function. For example, in seminal work from the Bissell laboratory, human breast epithelial cells were shown to abnormally proliferate (similar to tumor cells) when cultured as a two-dimensional monolayer, but displayed normal growth and formed structures typical of in vivo breast tissue when cultured in reconstituted basement membrane-based hydrogels[11]. Thus, recapitulating critical matrix cues is a promising approach to maintain salivary gland function. Indeed, freshly prepared salivary gland slices maintain acinar cell clusters that retain polarized morphology, $Ca^{2+}$ signaling, and secretory function[12] while aggregates of isolated salivary gland tissues, also known as salispheres, have been shown to polarize into organized tissues akin to acinar structures[13–15]. However, cells cultured using these approaches lose secretory function rapidly (over ~24–48 h), precluding evaluation of secretory dysfunction in general, and particularly, as a consequence of radiation damage, which requires longer times to realize physiological dysfunction[16–18]. Three-dimensional hydrogel-based engineered extracellular matrices have been explored to further enhance the longitudinal function of salivary gland tissues. These matrices include natural[8,19–23] and synthetic biomaterials[24–29]. Despite the promise of several of these approaches, which support excellent viability and lumen formation and appropriate apico-basolateral protein expression akin to the functional acinus, the maintenance of secretory function is elusive[19,20,23,24,26–38]. Hence, a major challenge for successful in vitro culture of salivary gland tissue is preserving acinar cell phenotype and function.

Our previous work established that dissociation of salivary glands to single cells using established methods[13] followed by hydrogel encapsulation resulted in low cell viability[28]. Viability was increased if cells were allowed to aggregate into spheres prior to encapsulation in hydrogels[28]. More recently, we demonstrated that encapsulation within matrix metalloproteinase (MMP)-degradable PEG hydrogels, which are degradable via cell-dictated processes, compared to hydrolytically degradable counterparts, promoted polarized lumen formation and the expression of acinar cell-specific markers, such as aquaporin 5 (Aqp5) and the sodium–potassium-chloride cotransporter (Nkcc1)[27]. However, expression of the Mist1 (bHLH15a) transcription factor, which is required to specify the acinar cell phenotype[39], decreased >99% by 2 days post-isolation[27], similar to other reports[19,20,23,24,26–29]. Further, interrogation of the fate of acinar cells post-encapsulation also revealed a transition to a keratin-expressing duct-like phenotype[26]. These studies confirm that further optimization of culture conditions is needed to support acinar cells and maintain functional tissues.

In this work, we sought to address the limitations of primary culture systems in the in vitro maintenance of secretory acinar cells to improve functional outcomes and increase experimental throughput. We synergized the supportive microenvironment provided by MMP-degradable PEG hydrogels with microbubble (MB) array technology[40,41] to assemble a modular salivary gland tissue chip platform. MB arrays have been successfully used as a convenient platform for culture of diverse cell types but this study is the first to use this platform for primary exocrine gland tissues[40,42–46]. The spherical architecture of the MB constrains tissue geometry to that of a functional acinus unit and provides a niche in which secreted factors are concentrated to maintain tissue viability and function[42,44]. MBs also allow long-term cell culture, and the ability to fabricate them in high-density arrays (>4000 MB/cm²) provides a useful platform for high-throughput and high-content screening[44]. In contrast to many tissue chip formats, this approach does not rely on stem cells or iPSCs, but on mature, fully differentiated salivary gland cell types, as protocols to generate salivary gland cells from iPSCs are still under development[47,48]. Conditions for isolation and in vitro culture of adult mouse and human salivary gland mimetics (SGm) were optimized and high-throughput functional assays were developed to assess calcium signaling and secretory cell function relative to native tissue. The resulting salivary gland tissue chips have been used to assay the response to radiation and the efficacy of a radioprotective agent. The culmination of this work is a tissue chip platform for both mechanistic studies and radioprotective drug screening to mitigate xerostomia.

## Results

**Partial tissue dissociation maintains acinus structure and cell–cell contacts**. To improve the maintenance of differentiated acinar cells in vitro, a modified dissociation protocol was adapted[8,21], which differs from previous isolation procedures[13,26–28]. Multicellular salivary gland tissue was isolated using sequential filtration to remove large tissue pieces (>100 µm), then blood cells and debris (<20 µm). The isolated acinar cell clusters and intercalated ducts (AIDUCs), which maintained characteristic morphology (Fig. 1a, b arrowheads), as well as fragmented striated ducts (Fig. 1a arrow) exhibited high viability (Supplementary Fig. 1) and were resuspended in culture medium. Immunofluorescent staining confirmed the maintenance of salivary gland epithelial markers within the cell clusters: AQP5, smooth muscle actin (SMA), cytokeratin 7 (K7), E-cadherin, Laminin, and MIST1 (Supplementary Fig. 2). In addition, polarization of cells within AIDUCs after isolation was demonstrated by basolateral staining of NKCC1 and inter-cellular staining of ZO-1 (Fig. 1b). Note that cell clusters seeded into the MB arrays include all cell types that may be present after isolation and size selection. This likely includes duct structures, such as striated ducts as seen in Supplementary Fig. 2c, which stain with K7, a luminal duct cell marker.

**Fabrication of MB array-based chips**. MBs are uniformly sized spherical cavities formed in polydimethylsiloxane (PDMS) by gas expansion molding using a templated array of deep cylindrical pits (Supplementary Fig. 3)[40,41]. Experiments in this study used MBs with spherical openings of 200 µm and maximum diameter of ~320 µm and arrays had a density of 278 MBs/cm² (Fig. 1c, d). Chips were affixed to the bottom of standard 48-well plates (1 chip/well), yielding ~107 MBs per well. The spherical architecture of the MB provides a unique niche enabling autocrine/paracrine conditioning, long-term culture, and transparency to enable in situ imaging for analyses[42,43,45].

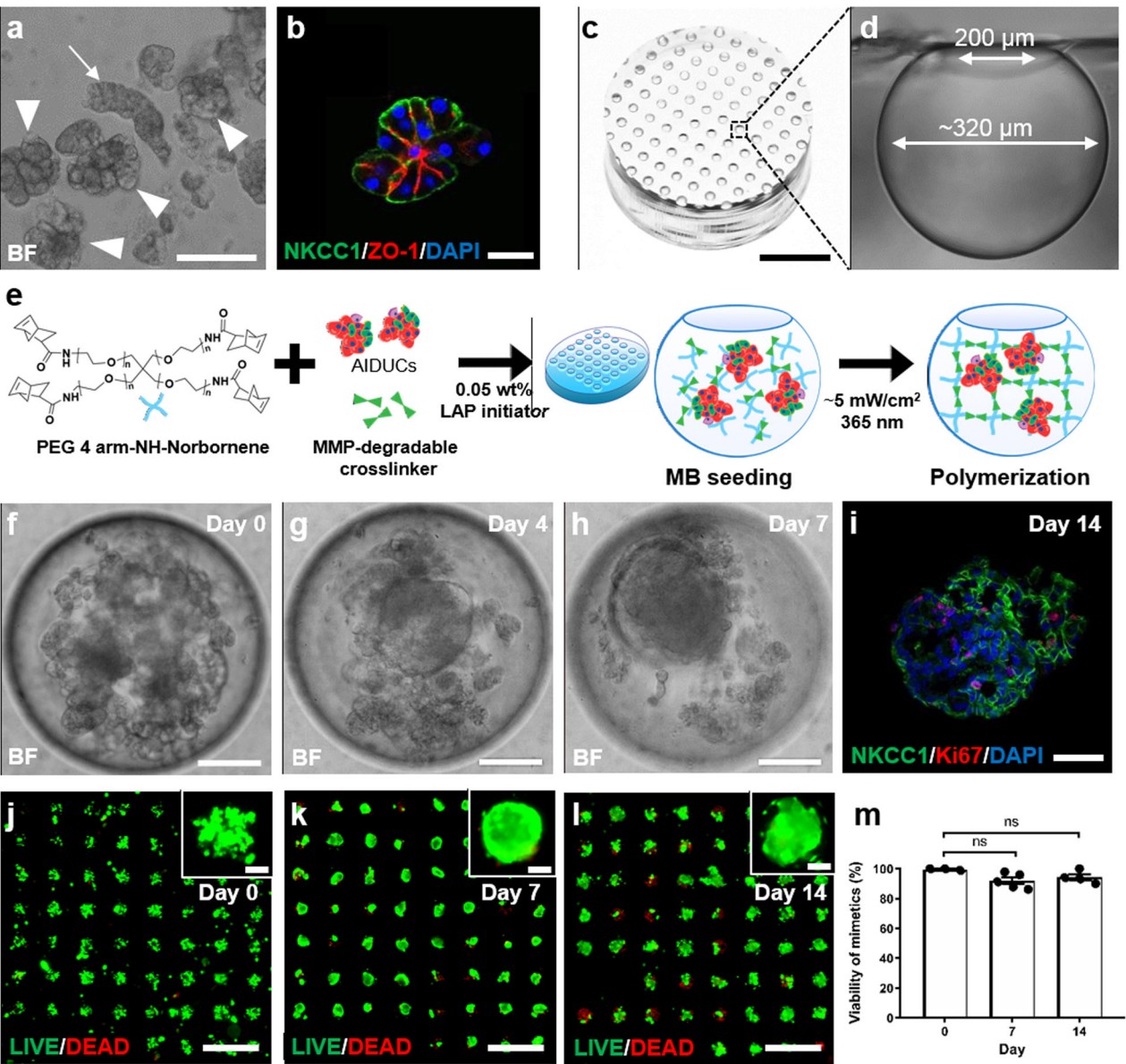

**Fig. 1 Characterization of salivary gland MB-hydrogel culture system. a** and **b** Representative images of **a** brightfield and **b** immunohistochemical staining of NKCC1 and ZO-1 in AIDUCs immediately post-isolation. Arrowheads in **a** indicate isolated acinar cell clusters. Arrow indicates a striated duct complex. **c** and **d** Image of **c** MB-chip (scale bar = 3 mm) and **d** a cross-sectional view of a single MB. **e** Schematic representation of hydrogel encapsulation of AIDUCs within MB-chips. **f**–**h** Brightfield images of AIDUC sphere formation in MB-chips at days 0, 4, and 7 (scale bars = 100 μm). **i** IHC staining for Ki67, NKCC1, and DAPI within AIDUCs at day 14 (scale bar = 40 μm). **j**–**l** Representative images of fluorescent LIVE (green) and DEAD (red) staining of AIDUCs in MB chips at (**j**) Day 0, (**k**) Day 7, and (**l**) Day 14 (scale bars = 1 mm, Insets = 100 μm). **m** Quantification of the percentage of MB containing viable cells at days 0, 7, and 14. Statistical analysis was performed using one-way ANOVA with Dunnett's post hoc test with α = 0.05. N = 3–5, n = 250–290 MB per time point. ns not significant.

**MB-hydrogel milieus promote cell aggregation and maintain cell viability in AIDUCs.** AIDUCs isolated from mouse submandibular glands (SMG) were mixed with precursor solution for MMP-degradable PEG hydrogels[26] (Fig. 1e), and pipetted onto MB arrays. After 15 min of incubation to allow settling of AIDUCs + gel precursor solution into the MBs, gels were polymerized, resulting in hydrogel-entrapped AIDUCs within the MBs (MB-hydrogels). From each SMG AIDUCs preparation, ~20 MB chips were prepared. Within the MB-hydrogels, brightfield imaging showed that the AIDUCs reorganized to form SGm by day 7 (Fig. 1f–h, Supplementary Video 1). Staining for the proliferation marker Ki67 indicated that cell proliferation continued in MB-hydrogels at least through day 14 (Fig. 1i). SGm

also exhibited high viability up to 14 days with >90% of the MBs within each array containing a majority of viable cells, as measured using LIVE/DEAD staining (Fig. 1j–m). Overall, these data indicate that maintenance of AIDUC cell–cell interactions and apical–basal polarity, including some basement membrane proteins, supports SGm survival and proliferation in MB-hydrogel.

**SGm formed in MB-hydrogels maintain acinar cell marker expression and polarization.** Gene expression was used to interrogate the phenotype of cells within MB-hydrogels. RNA was isolated from cultured SGm at days 4, 7, and 14 to determine expression levels using quantitative PCR of acinar cell-specific

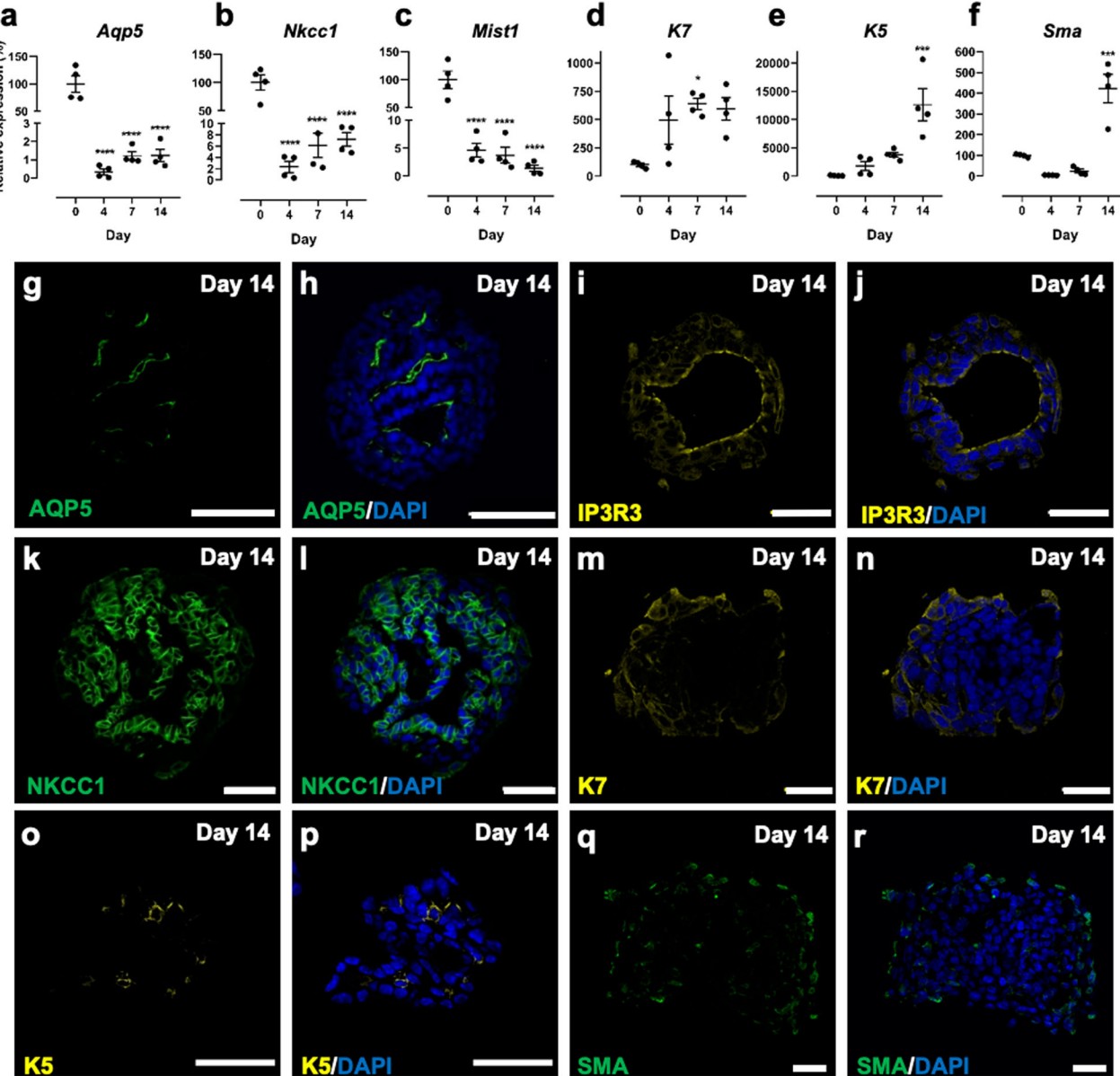

**Fig. 2 Mouse SGm in MB-hydrogels express markers of three major cell types in the salivary gland. a–f** Gene expression of *Aqp5* (**a**), *Nkcc1* (**b**), *Mist1* (**c**), K7 (**d**), K5 (**e**), and *Sma* (**f**) relative to day 0 with housekeeping gene *Rps29*. **g–r** Day 14 immunohistochemical staining of AQP5 (**g, h**), IP3R3 (**i, j**), NKCC1 (**k, l**), K7 (**m, n**), K5 (**o, p**), and SMA (**q, r**) in SGm co-stained with DAPI (**h, j, l, n–p, r** respectively). Scale bars = 40 μm. $N = 3$–6. Statistics are relative to Day 0, using ANOVA with Dunnett's post-hoc test. *$p < 0.05$, **$p < 0.01$, ****$p < 0.0001$.

markers *Aqp5*, *Nkcc1*, and *Mist1*. In comparison to AIDUCs immediately after isolation (day 0), the expression of all 3 markers was decreased in the SGm (Fig. 2a–c), consistent with previous work[26,27]. Nevertheless, expression was still detectable at day 14 with ~1.2%, 7.2%, and 1.4% of day 0 expression for *Aqp5*, *Nkcc*1, and *Mist1*, respectively. In contrast, SGm showed increases in gene expression at day 14 of the duct cell markers, cytokeratin 7 (*K7*, intercalated duct and luminal striated duct cells) and cytokeratin 5 (*K5*, luminal striated ducts), as well as the myoepithelial marker, smooth muscle actin (*Sma*) (Fig. 2d–f) to 5.95-, 126-, and 4.2-fold, respectively, above day 0 levels, likely due to the activation of K5 and K7 expression under stress/injury[26,49–52]. Initial SGm cultures were also investigated for expression of the neuronal marker βIII-tubulin, as nerves could be included with the AIDUC isolation. βIII-tubulin expression was undetectable, as indicated by observed high threshold cycles (see Supplementary

Fig. 4), suggesting that a neuronal population is not part of the SGm.

Immunohistochemical staining of SGm isolated at day 14 from MB-hydrogel showed that 99% of MBs developed internal lumens (Fig. 2g–r). Staining for AQP5 and inositol 1,4,5-triphosphate receptor 3 (IP3R3) showed these polarized acinar cell markers exhibited localized expression in cells lining the lumen (Fig. 2g–j), while NKCC1 staining (Fig. 2k, l) was observed throughout the SGm, consistent with intact gland expression patterns[27]. Staining for K7, K5, and SMA was limited to the exterior membranes of the SGm (Fig. 2m–r), consistent with duct and myoepithelial cell support of tissue function[53]. AQP5, IP3R3, NKCC1, K7, and SMA were also found to be present in SGm as early as day 7 (Supplementary Fig. 5). In addition, tight junctions and apicobasal polarization were maintained, as indicated by ZO-1 and E-Cadherin expression (Supplementary Figs. 5 and 6), as well

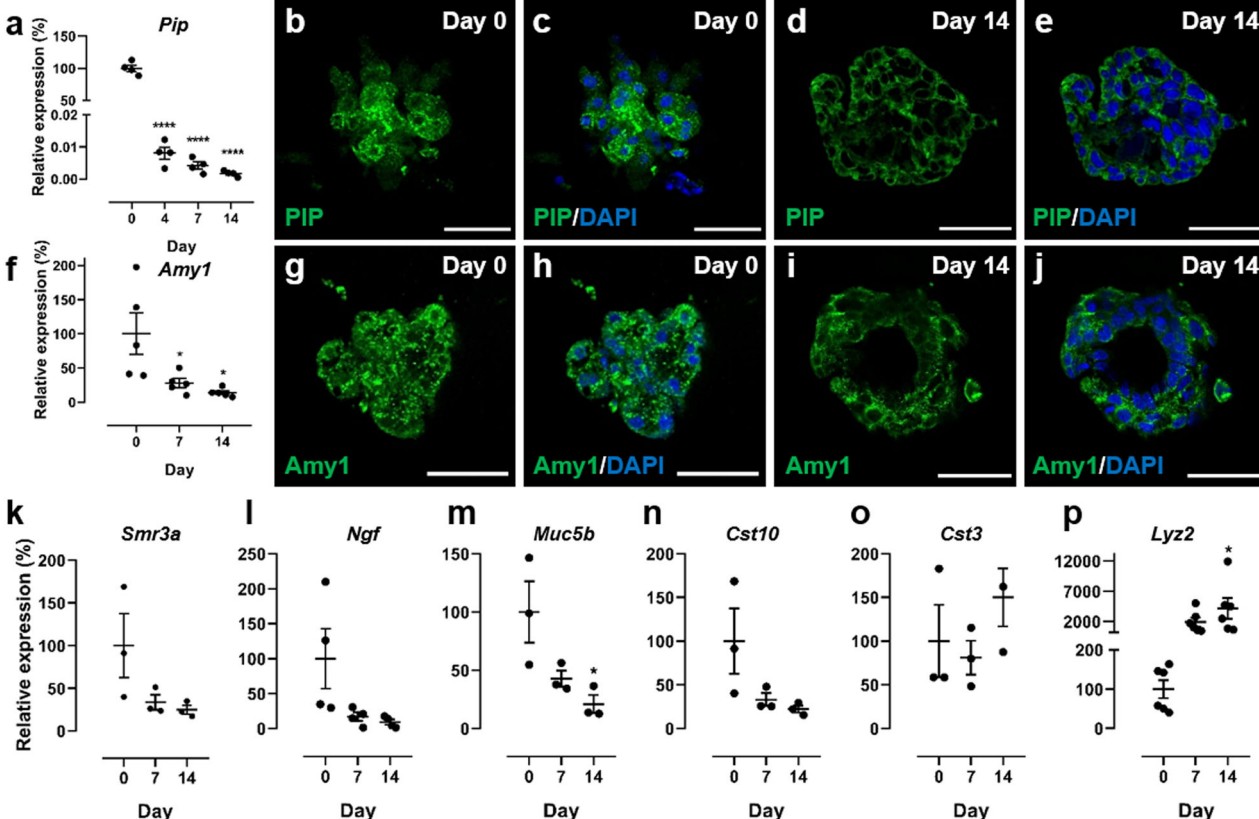

**Fig. 3 SGm in MB-hydrogels retain expression of secretory proteins. a** Gene expression of the secretory protein *Pip*. **b–e** PIP immunohistochemical staining at days 0 and 14. **f** Gene expression of amylase. **g–j** Amylase (Amy1) immunohistochemical staining at days 0 and 14. Scale bars = 40 μm. **k–p** Gene expression of *Smr3a* (**k**), *Ngf* (**l**), *Muc5b* (**m**), *Cst10* (**n**), *Cst3* (**o**), and *Lyz2* (**p**) determined using qPCR. All gene expression data is relative to day 0 with housekeeping gene *Rps29*. Statistics were performed using ANOVA with Dunnett's post-hoc test compared to day 0. *$p < 0.05$; **$p < 0.01$; ***$p < 0.001$; ****$p < 0.0001$; $N = 3–6$.

as tissue morphology, as indicated by Periodic acid-Schiff's-Alcian Blue (PAS-AB) staining in SGm (Supplementary Fig. 7). Interestingly, increases in gene expression for K5 and K7 are not corroborated by immunostaining (Fig. 2d, e vs. m & n, o & p), which shows modest K5 and K7 SGm staining. Underlying this result may be cellular outgrowth from the MB that is observed after day 7 and is positive for K5 and K7 (see Supplementary Fig. 8). As the outgrowth is not separated from the SGm for RNA isolation, outgrowth cells may be overwhelming the gene expression data for K5 and K7, leading to the observed discrepancy with immunostaining.

**SGm in MB-hydrogels maintain expression of secretory proteins**. To confirm that acinar cells retain function, the expression of secretory proteins was analyzed. qPCR and immunohistochemical staining were used to detect prolactin-inducible protein (Pip) and amylase (Amy1). Despite a drop in gene expression by day 14 relative to primary AIDUCs (Fig. 3a, f), immunostaining shows consistent expression of Pip (Fig. 3b–e) and amylase (Fig. 3g–j) in SGm between days 0 and 14, confirming that secretory marker expression is maintained in SGm.

While Pip and amylase are commonly used markers of secretory function in the salivary gland, a more comprehensive profile of secretory proteins expressed by SGm was also investigated. Specifically, gene expression was analyzed for proteins that are highly expressed in the SMG, selected using a published RNA-Seq dataset[54], microarray analysis data[55], and the Human Salivary Proteome Wiki (https://salivaryproteome.nidcr.nih.gov/public/index.php/Main_Page). Expression levels of the submaxillary gland androgen regulated protein 3a (Smr3a, Fig. 3k), nerve growth factor (Ngf, Fig. 3l), mucin 5b (Muc5b, Fig. 3m), cystatin 10 (Cst10, Fig. 3n), cystatin 3 (Cst3, Fig. 3o), and lysozyme 2 (Lyz2, Fig. 3p) were measured using qPCR. Lyz2, Cst3, Cst10, Muc5b, and Smr3a have roles in defense against oral bacteria and Ngf is a neurotrophin. Expression of Cst10, Muc5b, Ngf, and Smr3a decreased to ~9-25% of initial expression levels by day 14, consistent with other analyzed secretory products, and with the reduction in Mist1 in vitro. While Cst3 was statistically unchanged over 14 days, Lyz2 was upregulated to 41-fold at day 14 compared to day 0 tissue, which may implicate a hypoxia or endoplasmic reticulum-related stress response[56,57]. Overall, the expression analysis indicates that SGm maintain functionality, but the secretory phenotype declines over time relative to freshly isolated tissue, which may result from stress responses due to in vitro culture and the absence of supporting tissues (e.g., vasculature, neural, mesenchyme).

**Muscarinic and purinergic receptors are expressed in SGm**. The muscarinic acetylcholine receptor 3 (M3R) is the major neurotransmitter receptor involved in salivary fluid secretion from acinar cells. Acinar cells also express several types of purinergic receptors, including P2Y$_2$, a G protein-coupled-receptor, and P2X$_7$, a ligand-gated ion channel[58]. Activation of P2X and P2Y receptors leads to an increase in intracellular [Ca$^{2+}$] and can induce fluid secretion[59–61]. The expression of these receptors in SGm was interrogated using qPCR. At days 4, 7, and 14, $M_3r$ expression was decreased to 50%, 60%, and 14% of primary AIDUCs (Fig. 4a). $P2x_7$ expression decreased to 17% at day 4

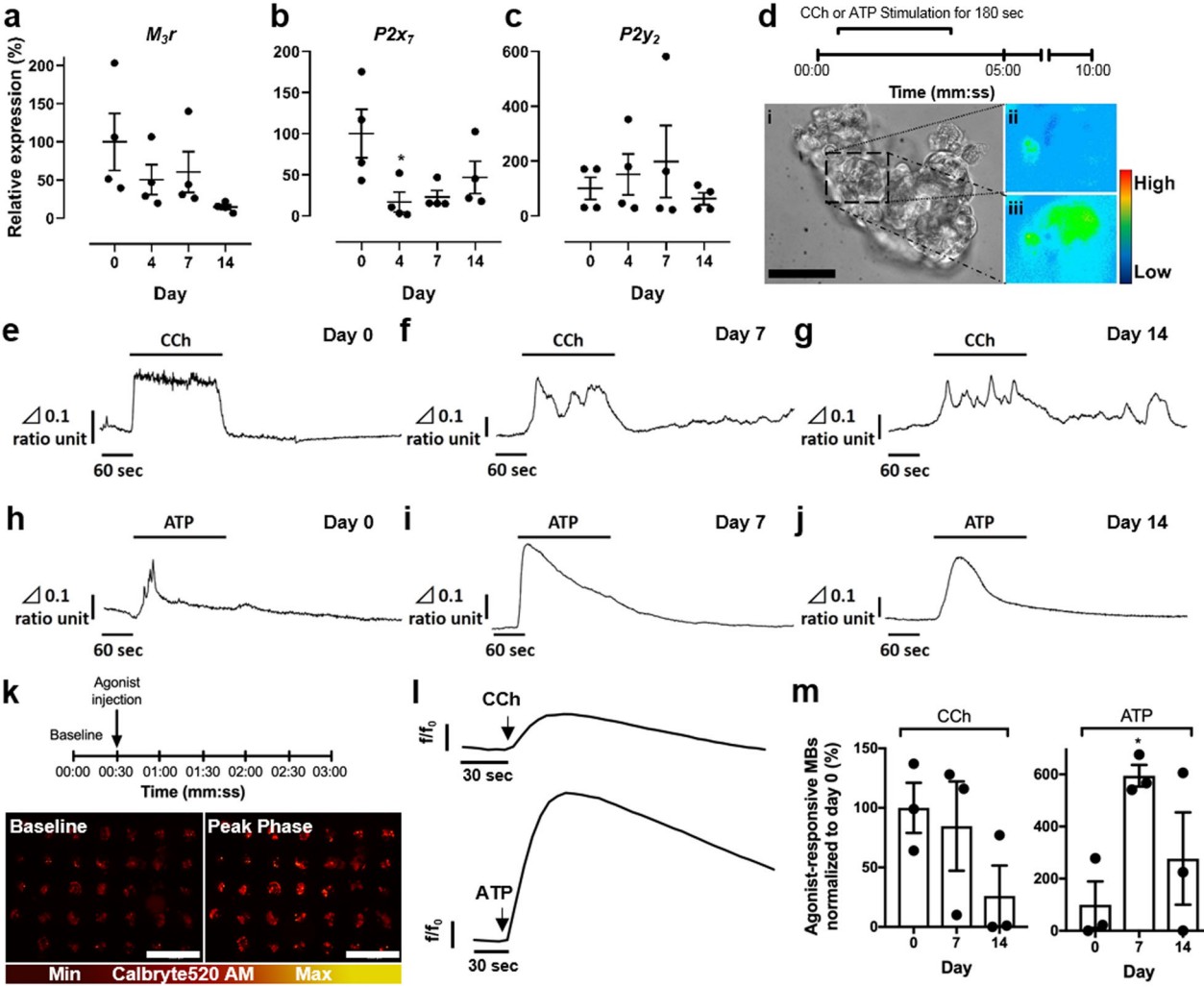

**Fig. 4 Mouse SGm in MB-hydrogels are responsive to stimulation with muscarinic agonist carbachol (CCh) and purinergic agonist ATP. a–c** Gene expression profiles of muscarinic receptor type 3 ($M_3r$: **a**) and purinergic receptors ($P2x_7$: **b** and $P2y_2$: **c**) in freshly isolated mouse salivary gland tissue and SGm at days 4, 7, and 14. **d** Experimental time-course of calcium imaging on mouse SGm using Fura-2 AM: fluorescent imaging was initiated at time 0 with continuous perfusion of buffer. CCh was then perfused for 180 s. After perfusion of buffer for 10 min, ATP was perfused for 180 s. Images of SGm at day 7: brightfield (**d**: i); baseline fluorescent intensity of $[Ca^{2+}]_i$ (**d**: ii) and peak phase fluorescence of $[Ca^{2+}]_i$ after stimulation with CCh (**d**: iii). Pseudocolor bar indicates relative intensity of $\Delta 340/380$ ratio. $\Delta 0.1$ ratio unit equals $340/380$ ratio $= 0.1$. Scale bar $= 50$ μm. **e–g** CCh stimulation leads to an elevation of $[Ca^{2+}]_i$ at day 0 (**e**) and in SGm (day 7: **f**, day 14: **g**). **h–j** Changes in $[Ca^{2+}]_i$ in AIDUCs (**h**) and in SGm (at day 7: **i**, day 14: **j**) are generated by stimulation with ATP. Vertical bar shows $\Delta 340/380$ ratio unit, and horizontal bar indicates time unit. **k–m** $Ca^{2+}$ signaling assay of SGm in whole tissue chips. **k** Experimental timeline of fluorescent imaging of $Ca^{2+}$-binding indicator Calbryte520 AM: baseline signals were collected in buffer from time 0, and CCh or ATP were injected at 30 s. **l** Representative fluorescent traces of responsive SGm at day 7 upon CCh and ATP stimulation. Data are represented as fluorescent intensity ($f$) divided by fluorescence at time 0 ($f_o$). **m** Percent of SGm in MBs responsive to CCh and ATP. Response is characterized by a significant difference between baseline timepoints and agonist timepoints via unpaired $t$-tests corrected for multiple comparisons using Holm–Sidak with $\alpha = 0.05$. Data is graphed and normalized to day 0 as 100%. Data showed mean ± SEM, $N \geq 3$, $n = 85$–97 per timepoint. $^{**}p < 0.01$, $^*p < 0.05$.

versus day 0 but increased to 47% by day 14 (Fig. 4b). $P2y_2$ expression shows trends of increasing to Day 7 then decreasing between Days 7 and 14 but due to significant variability in the data, expression is not statistically different at any timepoint versus Day 0 (Fig. 4c). Continued expression of the receptors responsible for propagating stimulatory signals supports the evidence that SGm maintain secretory function. It is interesting that $P2y_2$ expression is maintained at levels comparable to freshly isolated AIDUCs. While P2Y2 receptors have been implicated in the pathogenesis of diseases and inflammation[62], up-regulation of the receptor has been shown to confer protection in tissues under oxidative stress conditions[63,64]. Studies of mouse SMG cells cultured in vitro report up-regulation of $P2y_2$, which is important for migration, aggregation, and self-organization[59].

**Agonists stimulate intracellular calcium release in SGm.** Saliva secretion is dependent on intracellular calcium signaling. In functional SGm, treatment with an agonist, such as carbachol (CCh) or ATP, should stimulate a rapid increase of intracellular calcium $[Ca^{2+}]_i$, which drives fluid secretion from secretory acinar cells. Primary AIDUCs and SGm removed from MB-hydrogels were tested for CCh and ATP responsiveness using calcium flux analysis. After loading with the calcium-sensitive dye Fura-2, CCh and ATP were sequentially added to the perfusion imaging system; first CCh for 180 s within a 10-min interval; the system was then washed for 10 min with imaging buffer, and ATP was added for a 180 s incubation (Fig. 4d). Changes in $[Ca^{2+}]_i$ were determined by calculating ratios of 340/380 nm (Fig. 4e–j), as previously described[65,66]. An increase in $[Ca^{2+}]_i$ after stimulation with either

CCh (Fig. 4e–g) or ATP (Fig. 4h–j) was observed. Notably, at both 7 and 14 days of culture, SGm showed a response to CCh and to ATP with spatially distinct cells responding to each agonist (Supplementary Video 2). These results suggest that SGm within MB-hydrogel exhibit intact signaling pathways, as measured by agonist-stimulated calcium release.

Together with the expression of the $M3r$, $P2x_7$, and $P2y_2$ receptors and agonist-mediated $[Ca^{2+}]_i$ flux measured in individual SGm from MB-hydrogels, the results suggest that functional secretory cells are maintained for up to 14 days. However, common $[Ca^{2+}]_i$ flux methods are inefficient for screening purposes. Therefore, a high-throughput method for simultaneously detecting and measuring stimulated calcium flux of SGm in situ was developed. The chips were loaded with the calcium indicator Calbryte 520 AM at days 0, 7, and 14 of culture. CCh or ATP was added to the buffer and fluorescence was analyzed using time-lapse microscopy to record changes in calcium flux from each MB in the array (Fig. 4k–m, Supplementary Video 3). An overall response to CCh and ATP (Fig. 4l, m) was observed at days 7 and 14. Though the response trended toward lower in a time-dependent manner, stimulation of the SGm by CCh at both time points was statistically similar to AIDUCs at day 0 (Fig. 4m). Interestingly, a statistically significant ($p < 0.05$) increase was measured in the response to ATP over time, which is consistent with the maintenance of purinergic receptor gene expression (Fig. 4b, c) and previous reports[59]. The number of MBs in the array exhibiting ATP responsiveness at day 7 increased ~600% from day 0 (Fig. 4m). However, at day 14, the percentage of ATP-responsive MBs was not statistically different from day 0. To further validate function, SGm were also stimulated with the β-adrenergic agonist, isoproterenol (IPN), after verification of receptor expression (Supplementary Fig. 9a). An increase of $[Ca^{2+}]_i$ after stimulation with IPN (Supplementary Fig. 9b, c) was observed, which is consistent with receptor expression and activity observed with CCh and ATP stimuli. Together, these results demonstrate that individual MBs can be monitored longitudinally in an in situ array format, enabling high-throughput drug studies. Furthermore, the data validate the functional state of SGm over 14 days, albeit with waning function, as indicated by reduced CCh-stimulated signaling.

**Isolated human salivary gland tissues form viable SGm.** The successful generation of a functional salivary gland tissue chip using mouse tissue motivated efforts to develop a salivary gland chip using human tissue counterparts. Freshly isolated parotid gland tissue was obtained after informed consent from patients undergoing parotidectomy for benign disease. Samples used for these experiments were remote for any pathology. AIDUCs were prepared and encapsulated in gels within MBs. Chips of human parotid SGm were analyzed with LIVE/DEAD staining and showed cell viability of >82% across the array after encapsulation for up to 14 days (Fig. 5a–d), statistically equivalent to levels of cell viability observed for mouse SGm.

**Human SGm maintain expression of acinar cell-specific markers.** Gene expression analysis after 14 days of culture revealed that, similar to mouse SGm, human SGm expression of $AQP5$, $NKCC1$, and $MIST1$ was significantly ($p < 0.05$) reduced by day 2 (Fig. 5e–g). However, expression of both $NKCC1$ and $MIST1$ was maintained at 19% and 13% up to day 7, which is substantially higher than previously published levels[27], indicating that the AIDUC isolation approach combined with the MB-hydrogel microenvironment significantly improved longitudinal tissue function in vitro. Also as observed for mouse cells, human SGm

expressed the muscarinic ($M3R$) and purinergic ($P2X7$, and $P2Y7$) receptors (Fig. 5h–j). In contrast to mouse SGm, all three receptors were reduced in human parotid SGm over time compared to isolated primary human AIDUCs. Nevertheless, expression remained detectable even at day 14, at ~5%, 28%, and 26% of day 0 tissue for $M3R$, $P2X7$, and, $P2Y2$, respectively.

Immunofluorescent labeling after 14 days revealed that, similar to mouse SGm, the human SGm formed internal lumens and exhibited polarized localization of the acinar protein markers NKCC1 and AQP5 (Fig. 5k–m). The duct cell marker K7 is localized to the outside of the SGm (Fig. 5k), the tight junction protein ZO-1 is expressed at both the luminal and outer surfaces (Fig. 5l), and the acinar-specific water channel AQP5 is expressed at the luminal surface (Fig. 5m). Thus, human parotid SGm encapsulated in MB-hydrogels maintain expression of polarized and functional acinar cell markers in vitro for up to 14 days. Furthermore, secretion of amylase from human SGm within the MB-hydrogels was detected in >86% of MBs at day 7, and >68% by day 14 of culture (Fig. 5n) (raw, unnormalized data, Supplementary Fig. 10).

**Human SGm respond to calcium agonists.** To determine if human SGm remain functional within the MB-hydrogels, individual SGm were isolated at day 0, 7, and 14, and used for imaging $[Ca^{2+}]_i$ after stimulation with CCh and ATP. CCh elicited a marked elevation in the 340/380 fluorescence ratio from day 7 and 14 (Fig. 5o–q), although SGm at day 14 showed a delayed response. Stimulation of purinergic receptors with ATP showed an increase in $[Ca^{2+}]_I$ at both day 7 and 14 that was as robust as in primary AIDUCs (Fig. 5r–t), and imaging demonstrated that individual cells within the human SGm exhibit these responses (Supplementary Video 4).

High throughput screening of human SGm was conducted as described, using the Calbryte 520 AM fluorescent marker. Chips were first stimulated with CCh and subsequently with ATP (Fig. 5u–w, Supplementary Video 5). $[Ca^{2+}]_i$ was increased upon stimulation at both day 7 and 14 of culture (Fig. 5w). Consistently, the response to CCh and ATP paralleled gene expression levels of the $M3R$ and $P2Y2$ receptors, which also decreased from day 0 to day 7 (Fig. 5h, j). The elevated response to ATP at day 14 was again consistent with higher expression of the P2X7 receptor in the cultured cells. The calcium flux analysis was quantified and normalized to day 0, showing an average response for muscarinic stimulation of 80% and purinergic stimulation of 160%, relative to primary AIDUCs (Fig. 5w). These data indicate that human SGm retain functional activity in MB-hydrogel tissue chips and may be useful for drug screening.

**Murine SGm recapitulate in vivo responses to radiation and radioprotection.** To demonstrate the potential of the salivary gland tissue chips for drug screening, the response of murine SGm to singular and fractionated radiation alone, and after treatment with the radioprotective agent, WR-1065, was analyzed (Supplementary Fig. 11). First, the MB-hydrogel mouse SGm were validated against a common murine in vivo irradiation model that emulates radiation-induced xerostomia in humans[17,67]. Mouse AIDUCs were seeded into MB-hydrogels in chips. After 4 days in culture, chips were irradiated with 0, 7.5, 10, or 15 Gy. After 2 days, individual SGm were removed from MB-hydrogel and immunostained for the DNA damage marker γH2AX. Staining was compared to in vivo murine SMG irradiated at 15 Gy 2 days prior to isolation. Quantification of staining (Fig. 6a) shows a dose-dependent increase in γH2AX foci/nuclei from 1 to 3.3 within SGm. A dose of 15 Gy shows statistical equivalence ($p = 0.78$) to the murine irradiation model,

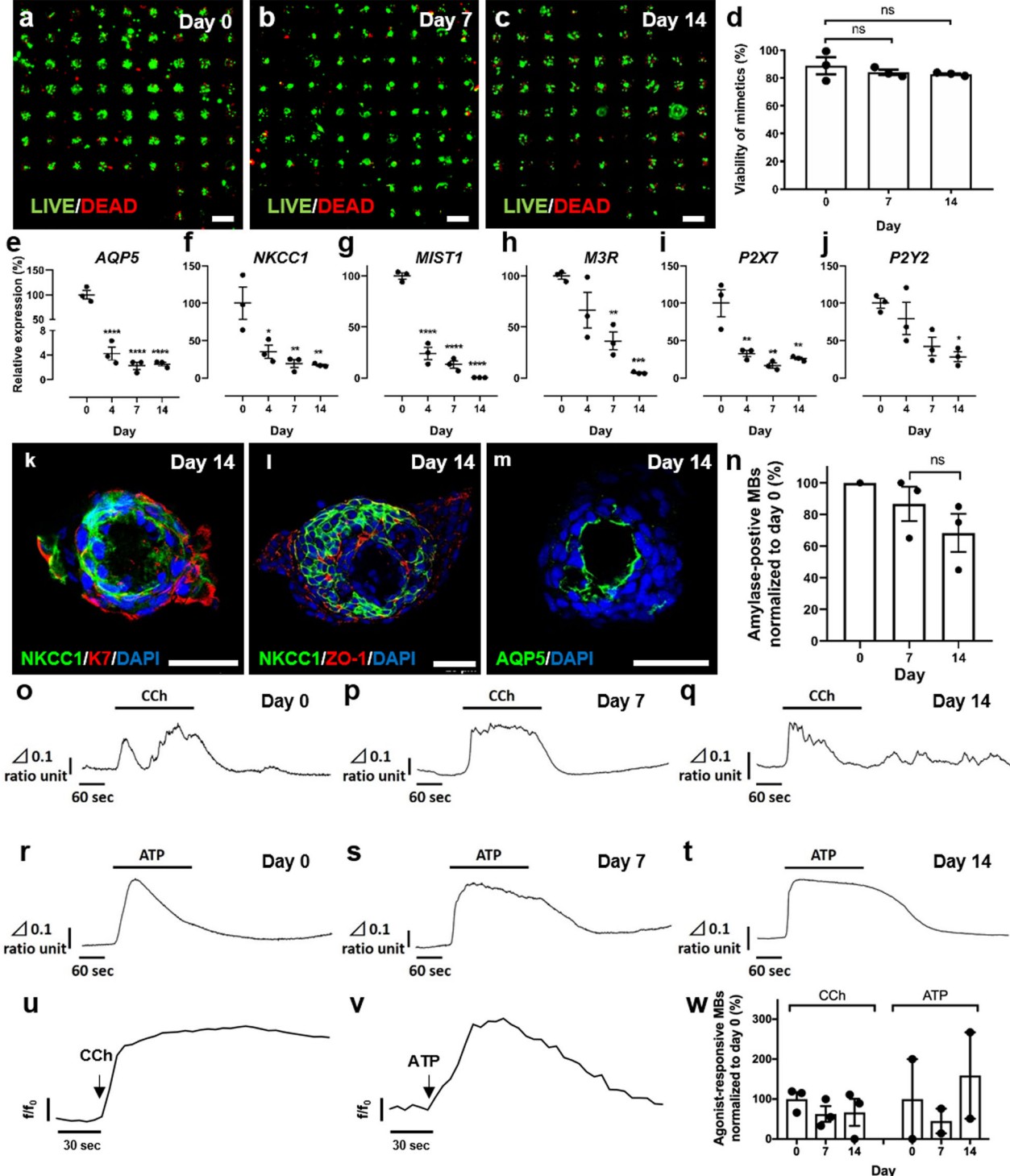

which results in ~3.1 γH2AX foci/nuclei 2 days after irradiation. Therefore, a singular dose of 15 Gy or the fractionated equivalent (e.g., 5 sequential days of 6 Gy) were used for subsequent studies. Two days after irradiation with the singular 15 Gy dose, SGm viability and DNA damage were evaluated. LIVE/DEAD staining showed no change in irradiated SGm 2 days post-irradiation or in SGm irradiated with addition of WR-1065 (Fig. 6b–d). Quantification of the images showed >90% of the MBs in the array contain a majority of viable cells in the three groups (Fig. 6e). SGm were immunostained for the DNA damage markers γH2AX (Fig. 6f–j) and 53BP1 (Fig. 6k–o). Quantification showed a 3-fold

increase in γH2AX-positive foci (Fig. 6p) and a 9-fold increase in 53BP1-positive foci (Fig. 6q) in SGm isolated from irradiated chips (single and fractionated), in comparison to unirradiated control chips and to previous reports of irradiated glands[17,68]. Quantification also demonstrated that the radioprotectant WR1065 reduced the number of γH2AX and 53BP1 foci by ~50% in comparison to chips that received only irradiation irrespective of the radiation regimen (Fig. 6p, q). These results provide evidence that the salivary tissue chips offer a functional and responsive platform for future use in studies of radiation damage and radioprotection of salivary glands.

**Fig. 5 Human SGm in MB-hydrogels maintain viability and retain markers of functional phenotype. a–d** LIVE/DEAD assay of human chips at days 0 (**a**), 7 (**b**), and 14 (**c**), and quantification (**d**). Scale bar = 500 μm. Statistical analysis was performed using one-way ANOVA with Dunnett's post hoc test. **e–g** Gene expression of acinar markers *MIST1* (**e**), *NKCC1* (**f**), and *AQP5* (**g**) measured using qPCR. **h–j** Gene expression of muscarinic receptor *M3R* (**h**), and purinergic receptors *P2X7* (**i**), and *P2Y2* (**j**). **k–m** immunofluorescent staining of SGm at day 14 with acinar markers; NKCC1 (**k, l**, green), AQP5 (**m**, green), ductal marker K7 (**k**, red), and tight junction ZO1 (**l**, red). Scale bar = 40 μm. **n** Amylase expression per MB at day 0, day 7, and day 14, normalized to day 0 levels. $N = 3$ and $n = 45–97$ per $N$. Raw fluorescence data for the normalized data can be found in Fig. S7. **o–t** Calcium signaling analysis of AIDUCs at day 0 and SGm at day 7 and 14 from MB-hydrogels. CCh and ATP stimulation leads to an elevation of $[Ca^{2+}]_i$ in AIDUCs (CCh, **o**; ATP, **r**) and of $[Ca2+]_i$ in SGm in MB-hydrogels at day 7 (CCh, **p**; ATP, **s**) and, day 14 (CCh, **q**; ATP, **t**). Vertical bar shows Δ340/380 ratio unit, and horizontal bar indicates time unit. **u–w** Whole chip calcium assay via fluorescently labeled calcium-binding indicator Calbryte 520 AM. **u** and **v** Representative trace of SGm response to CCh (**u**) and ATP (**v**) stimulation at day 7. **w** Quantification of percent responsive microbubbles per chip. Data are represented as fluorescent intensity ($f$) divided by fluorescence at time 0 ($f_o$). Percent of SGm in MBs responsive to CCh and ATP (**w**). Response is characterized by a significant difference between baseline timepoints and agonist timepoints via unpaired $t$-tests corrected for multiple comparisons using Holm–Sidak with $\alpha = 0.05$. Data are mean ± standard error of measurement, $N \geq 3$, \*\*$p < 0.01$, \*$p < 0.05$ except **w**, where $N = 2–3$ and $n = 45–97$ per $N$. ns not significant.

## Discussion

This study reports on the use of engineered extracellular matrices combined with MB arrays to enable longitudinal maintenance of functional SGm in an arrayed tissue chip format. The ex vivo tissue mimetics were validated, using both mouse and human tissues, for functional similarities to freshly isolated salivary tissue. Immunostaining indicates SGm contain all major cell types of the salivary gland including acinar, duct, and myoepithelial cells. Although gene expression of acinar markers is reduced over the 14-day culture period, there is detectable expression of important functional secretory markers. SGm maintain secretory function, as measured using immunostaining of secretory proteins, and secretory agonist-mediated $Ca^{2+}$-flux. Moreover, SGm were used in proof-of-principle testing of a known radioprotective compound, WR1065, the active metabolite of amifostine. Amifostine has been established as an effective agent for radioprotection of salivary glands and is FDA-approved for clinical use. However, this therapy is accompanied by undesirable side effects, motivating the need to identify alternative radioprotective agents. Obstacles to finding such drugs include the time and expense of animal screening and testing. The salivary gland tissue chip provides a validated testing platform, allowing screening of drug libraries while requiring only a fraction of the animal testing, which would otherwise be used.

The salivary gland tissue chip was developed using mild dissociation of mouse or human salivary gland tissues, resulting in a heterogeneous population of gland cells (e.g., ductal cells, acinar cells, and possibly myoepithelial cells) for SGm development. This approach sidestepped the challenges associated with differentiation of embryonic or induced pluripotent stem cells (ESC, iPSC) to recapitulate salivary gland tissue organization and function. The use of ES or iPS cells is a common and highly successful approach in the development of other ex vivo tissue mimetics or tissues-on-a-chip[69–77]. While progress has been made in driving the differentiation of ES cells toward an embryonic-stage salivary gland, the resulting tissue remained immature in vitro and only showed maturation when orthotopically implanted into mouse[47,48]. Similarly, transdifferentiation of MSCs has been reported to generate acinar-like salivary gland cells[78], but the molecular pathways involved are unknown. Steps toward iPSC differentiation into salivary gland tissue are still unclear. Thus, the lack of a defined protocol for programming iPS or ES cells to generate SGm led us to pioneer the use of AIDUCs in this in vitro platform. The advantage of maintaining the salivary structures as functional units for in vitro culture has recently been noted by others[79]. The AIDUC approach also mitigates the risk of underrepresenting cell types present in the adult gland, which may be derived from separate differentiation programs,

and may be critical for proper function of this tissue mimetic system.

While the dissociation method used improves upon more rigorous and stressful approaches, the native tissue microenvironment is still disrupted. Removal of supportive tissue and matrix structures may contribute to the reduction in secretory function observed over the 14 days analyzed, which is supported by upregulation of stress markers, including ATP responsiveness, $P2X_7$, and lysozyme[80]. Stress-related reduction in function is consistent with similar tissue mimetics including the mammary, pancreas, and parathyroid glands[81]. Here, a portion of the basement membrane and other critical matrix components are stripped from the AIDUCs. To recapitulate some of these matrix cues, a PEG-based hydrogel was utilized to support tissue function. Hydrogels have been used in a variety of tissue mimetic approaches from fundamental to drug screening and discovery studies[82–87], including development of tissue organoids[88,89], neural tube structures[90,91], and generation of blood–brain barrier models[92]. The hydrogels employed here have very modest biofunctionalization and are formed from oft-used, nonspecific MMP-degradable crosslinkers[28,93–100]. Despite the fairly rudimentary design of this engineered extracellular microenvironment, it showed great promise as a platform to support salivary tissue structure and function. Previous studies by us and others have shown that various hydrogels based on Matrigel, hyaluronic acid, decellularized extracellular matrix, fibrin, and poly(ethylene glycol) can improve expression of acinar markers including amylase, Aqp5, Mist1, and Nkcc1 compared with traditional 2D culture[19,27,32–38]. Specific to the work here, PEG hydrogels offer flexibility to introduce more complex matrix cues to enhance tissue development and function[101–103]. For salivary gland, laminin, collagen, and fibronectin can be incorporated by simple entrapment[103] or short, bioactive peptides can be covalently linked into hydrogel networks[94,103,104]. Indeed, incorporation of the laminin-111 peptide, especially in a trimeric presentation, influences salivary gland cell organization and function in vitro and potentiates gland regeneration in vivo[37,38]. Furthermore, proteolytic matrix remodeling has been shown to be critical for remodeling and apicobasal organization, and the rate at which the matrix can be degraded may have a significant impact on the organization and maintenance of the secretory phenotype of tissue mimetics[105–107]. Due to the highly tunable chemistry offered by the PEG hydrogel system, other crosslinkers can be easily incorporated to tune degradation[108–111], altogether seeking to improve the long-term phenotype and function of secretory acinar cells.

In addition, several supportive cell and tissue types are disrupted during dissociation and may be required to enhance

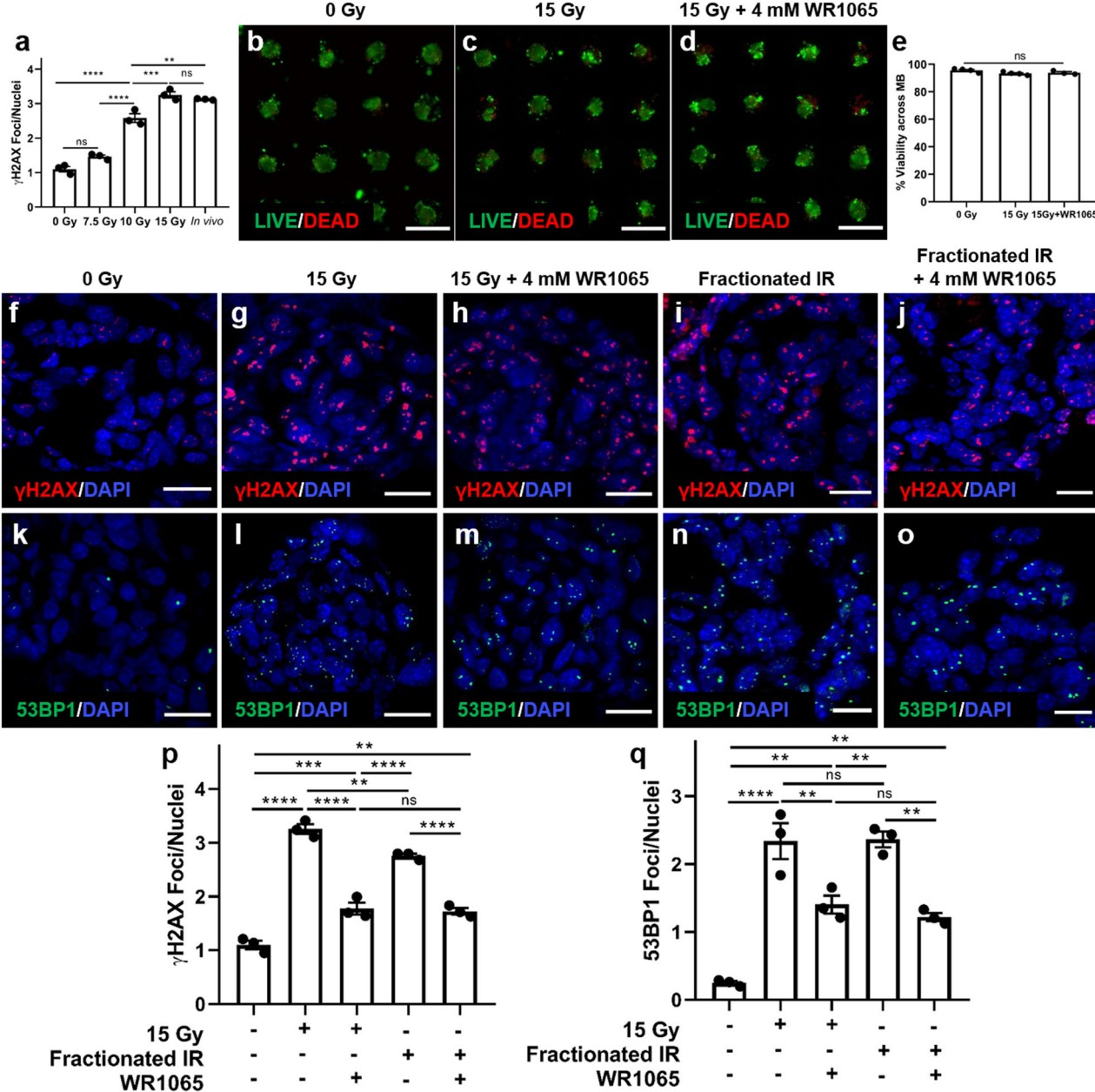

**Fig. 6 Treatment of Mouse SGm in MB-hydrogels with WR-1065 demonstrates radioprotection from DNA damage after irradiation. a** DNA damage response after irradiation of SGm in MB-hydrogels compared to in vivo response. **b–d** Cell viability of encapsulated SGm after irradiation with 0 and 15 Gy, and validation of radioprotective effect of WR1065. Scale bar = 500 μm. **e** Quantification of the percentage of MB containing viable cells at day 2 post-irradiation. **f–o** Effect of irradiation (1 × 15 Gy and equivalent fractionated doses) and WR1065 on DNA damage, as indicated by IHC staining of γH2AX (**f–j**) and 53BP1 (**k–o**). Scale bar = 15 μm. **p** and **q** Quantification of γH2AX and 53BP1 foci per cell. Statistical analysis was performed using Ordinary one-way ANOVA with Tukey's multiple comparisons test with $\alpha = 0.05$. $N = 3$, $*p < 0.05$; $**p < 0.01$; $***p < 0.001$; $****p < 0.0001$, ns not significant.

longitudinal SGm function. These tissues include vasculature, nerves, and mesenchyme, which are intrinsic to salivary gland development, homeostasis, and regeneration[13–15,112]. For example, primary embryonic mesenchyme supported salivary organoid formation with robust AQP5 expression in co-culture with E16 salivary epithelium[113,114]. Additionally, incorporation of parasympathetic ganglion (PSG) cells in cultured E13 murine SMG and sublingual glands or salivary gland salispheres promoted differentiated AQP5-positive acinar cells[22,115]. Removal or absence of the PSG in mouse SMG explant culture decreased branching morphogenesis and expression of basal progenitor cell markers[116,117]. Finally, inhibition of vascular endothelial growth factor receptor type 2 (VEGFR2) or depletion of endothelial cells

(ECs) from embryonic SMG led to loss of acinar cells, suggesting that ECs are important to support salivary function[118]. Thus, integration of more complex chip architectures enabling incorporation of these supportive cells may enhance the function and longevity of ex vivo SGm.

Soluble factors are commonly used within culture media of ex vivo tissues to provide a simple and controllable means to maintain or promote tissue function and emulate important supportive tissues lost during isolation. For example, the use of neurotrophic factors, such as neurturin and glial-derived growth factor (GDNF), may provide powerful signals due to their roles in salivary gland development and effectiveness in preventing stress responses, such as those occurring after irradiation

damage[116,119–122]. Additionally, Wnt agonists stimulate long-term expansion and self-renewal of salivary gland organoids[123]. Through a combination of Wnt activation, knockdown, and knock-in experiments, Shin et al. showed that Wnt3A and R-spondin induced the expression of stem cell markers, while knockdown of Wnt3A reduced expression of acinar cell markers[124]. FGFs, produced by the mesenchyme, provide important factors to maintain an acinar-supportive niche[125,126]. For example, FGF2, together with laminin-111, enhanced AQP5 expression in developing terminal buds[114]. FGF7 and FGF10 were also important for submandibular branching, with FGF7 enhancing bud expansion and FGF10 promoting ductal elongation[127]. Furthermore, Sui et al.[48] reported that FGF10 supplementation promotes correct spatial distribution of acinar and ductal structures, increases expression of acinar markers (Aqp5, Mist1, amylase), and enhances the organoid response to CCh stimulation. Additionally, the work by Adine et al. suggests FGF10 enhances the development of epithelial SG-like organoids[128]. While the media used in this study included FGF2, the addition of FGF7, FGF10, or other combinations of FGFs could further enhance tissue function. Several studies have also documented the effects of neural factors in promoting the branching and development of glandular tissues, although these factors may not be sufficient[129–131]. Additionally, the ROCK inhibitor Y-27632 has been used to reduce cell stress-related loss of secretory function in salivary gland cultures[14,106,113]. Alternatively, induction of acinar-to-ductal metaplasia or transdifferentiation via TGFβ inhibition[114,132], EGFR inhibition[22,133], or Src and p38 MAPK inhibition[134] may prevent acinar cell loss.

## Conclusions

We have developed tissue chip arrays that enable in vitro culture and screening of functional mouse and human SGm. SGm can be used to survey salivary gland radioprotective drugs in a high-throughput manner. This in vitro culture platform will enable large-scale studies, which are currently prohibitive due to the limitations of in vivo testing. One use for the salivary gland tissue chip is to investigate gland radiosensitivity and to screen for less toxic and more effective radioprotective drugs than those currently in use. In addition, the availability of functional in vitro SGm arrays is expected to provide a tool for mechanistic studies, as well as clinically predictive screening assays.

## Methods

**Animals.** Female C57BL/6J (Jackson Laboratory, ME, USA) mice at 4–12 weeks of age were used in this study. Animals were maintained on a 12-h light/dark cycle and group-housed with food and water available ad libitum. All procedures were approved and conducted in accordance with the University Committee on Animal Resources at the University of Rochester Medical Center.

**Human tissue.** Freshly isolated adult human parotid salivary gland tissue was collected from consenting male and female patients, average age = 61.8 years, undergoing parotidectomy for benign disease. Samples used for these experiments were remote for any pathology. This study was approved by the University of Rochester Research Subjects Review Board (RSRB00001326).

**Preparation of mouse AIDUCs.** Primary mouse SMG clusters were isolated using a modified protocol based on previous publications[8,135]. Briefly, mice were euthanized with $CO_2$ and cervical dislocation. SMGs were removed and completely minced with a razor blade. Minced tissue was suspended in 5 mL Hank's buffered salt solution (HBSS) with 2 mM $CaCl_2$ and 1 mM $MgCl_2$, 100 U/mL collagenase type II, 1 mg/mL hyaluronidase, 15 mM HEPES, and 0.5% bovine serum albumin, and incubated at 37 °C for 60 min in media continuously conditioned with 95% $O_2$ and 5% $CO_2$. Dispersed SMG clusters were subsequently passed through 20 and 100 μm mesh filters. The isolated clusters were re-suspended in culture medium [Dulbecco's modified Eagle medium (DMEM): F-12 (1:1) supplemented with 100 U/mL Penicillin and 100 μg/mL Streptomycin, 2 mM Glutamine, $N_2$ supplement (0.5 × Invitrogen), 2.6 ng/mL insulin, 2 nM dexamethasone, 20 ng/mL

epithelial growth factor (EGF), and 20 ng/mL basic fibroblast growth factor (bFGF)[13,28].

**Preparation of human AIDUCs.** Dissected human parotid tissue was placed into ice-cold PBS and stored on ice for transport to the laboratory, and was subsequently dissociated within 2 h after surgical removal. Dissociation of the human salivary gland tissue was done as described for mouse tissue, except the final concentrations of collagenase and hyaluronidase were doubled.

**Synthesis and characterization of hydrogel precursors.** 4-Arm PEG-Norbornene was synthesized by functionalization of 4-arm 20 kDa PEG-NH₂ with norbornene using N,N′-dicyclohexylcarbodiimide (DCC) coupling, as previously described[27,28,136]. The structure and percent functionalization (>90%) of lyophilized macromer was determined by ¹H-NMR: 4-arm PEG-amine norbornene (¹H NMR (CDCl₃): $d$ = 6.0–6.3 ppm (norbornene vinyl protons, 8H, multiplet), 3.5–3.9 ppm (PEG ether protons, 1817H, multiplet)).

The peptide GKK**C**GPQG↓IWGQ**C**KKG was synthesized by standard solid-phase peptide synthesis on FMOC-Gly-Wang resin (EMD) using a Liberty 1 Microwave-Assisted Peptide Synthesizer (CEM) with UV monitoring, as previously described[27,28,95]. The peptide was cleaved and deprotected using a cleavage cocktail composed of 92.5 vol% trifluoroacetic acid (TFA), 2.5 vol% triisopropylsilane, 2.5 vol% ddH₂O, and 2.5 vol% 3,6 dioxa-1,8-octane dithiol and mixing for 2 h. Cleaved peptide was separated from resin via vacuum filtration and precipitated in ice-cold diethyl ether and collected by centrifugation thrice. After drying overnight in vacuo, the peptide was dialyzed against ddH₂O using 500 MWCO dialysis tubing for 48 h and lyophilized. Peptide molecular weight was verified using a Bruker AutoflexIII Smartbeam matrix-assisted laser desorption ionization time-of-flight mass spectrometer using α-Cyano-4-hydroxycinnamic acid matrix dissolved in 50:50 acetonitrile: H₂O + 1% TFA. Bruker Peptide Calibration Standards were used for calibration. Peptide purity was analyzed via 205 nm absorbance measured in ddH₂O with an Evolution UV/Vis detector (Thermo Scientific) using a previously published methodology[137]. The photoinitiator, lithium phenyl-2,4,6-trimethylbenzoylphosphinate (LAP), was synthesized in a two-step process and validated using ¹H-NMR, as previously described[136].

**Fabrication of MB arrays.** Gas expansion molding was used to fabricate MB well arrays in PDMS, as described in Supplementary Fig. 3 and previously reported[40,41]. Briefly, a mixture of 10:1 (by wt) ratio of base to curing agent (Dow Corning Sylgard 184 PDMS) was poured over a silicon wafer template containing an array of etched cylindrical pits (200 μm diameter spaced 600 μm apart on a square lattice) and cured at 100 °C for 2 h. MB wells form uniformly over each pit. After curing, the cast was peeled from the template and chip arrays were cut to size using a circular punch. Chips were then glued into the wells of standard tissue culture plate using a drop of PDMS (5:1 wt% ratio of base to curing agent), which was allowed to cure overnight at room temperature. Chips were primed in a desktop vacuum chamber with 70% ethanol followed by copious washing with PBS prior to AIDUCs seeding.

**Seeding and culture of AIDUCs in MB-hydrogels.** The isolated AIDUCs were mixed with the hydrogel precursor solution comprised of 2 mM norbornene-functionalized 4-arm PEG-amine macromers and 4 mM of the dicysteine functionalized MMP degradable peptide (GKK**C**GPQG↓IWGQ**C**KKG), 0.05 wt% of the photoinitiator LAP, and 0.1 mg/mL laminin in DPBS at ~7.5 × 10⁵ cells/mL. The AIDUC/gel precursor solution was pipetted onto MB arrays. After 15 min, a RainX-treated glass coverslip was put onto the top of the MB chip to remove excessive AIDUCs + gel precursor and the gels were polymerized to entrap the AIDUCs within MB using 365 nm, ~5 mW/cm² light for 5 min. Then, 1 mL of culture media was added to each well and the glass coverslip was carefully detached. Culture medium was changed every 24 h.

**Measure of SGm viability.** At days 0, 2, 7, and 14 after MB seeding, calcein AM (2 μM) and ethidium homodimer (4 μM) were added directly to culture medium. After incubation for 30 min at 37 °C, samples were imaged with an FV1000 laser scanning confocal microscope (Olympus). The number of MB with calcein AM-positive live cells was quantified using ImageJ.

**RNA extraction and quantitative PCR.** MB-gel chips were rinsed with PBS and placed into 350 μL TRK lysis buffer (Omega Bio-tek) containing β-mercaptoethanol (β-ME, 20 μL per 1 mL lysis buffer) and stored at −80 °C until RNA extraction. Total RNA was extracted using the OMEGA kit (Omega Bio-tek) and reverse-transcribed using the iScript™ cDNA synthesis kit (Bio-Rad), according to the manufacturer's instructions. Quantitative PCR analysis of individual cDNAs was performed on a CFX96™ Real-Time System (Bio-Rad) using PowerUp SYBR™ Green Master Mix (Bio-Rad) for genes listed in Supplementary Table 1. Quantitative PCR results were normalized to mouse Rps29 and human RPS29 mRNA levels and analyzed using the $2^{-\Delta\Delta CT}$ method.

**Immunohistochemistry and imaging**. SGm were fixed with 4% PFA for 45 min at room temperature with shaking and washed three times with PBS. Due to the challenge of sectioning aggregates cultured in PDMS chips, a new process was developed to enable removal of tissue encapsulated within hydrogels from the tissue chips. The chips were incubated with xylene for 30 min to swell the PDMS resulting in significantly increased MB diameters. Then, cryogel was added onto the top of the MB arrays followed by flash freezing which enabled 'pop off' of the aggregates–gel complex from the MB in an array format for subsequent sectioning. After embedding in optimal cutting temperature solution (OCT, Tissue-Tek) and freezing at −20 °C, 10 μm sections were cut using a HM 550 (Micron) cryostat, which were mounted on SuperFrost Plus slides (Fisher Scientific). After antigen retrieval via incubation for 10 min in HIER buffer (10 mM Tris base, 1 mM EDTA solution, pH 9.0), sections were first washed three times in PBS for 5 min and blocked in 10 vol% normal donkey serum + 0.1 wt% bovine serum albumin (BSA) for 1 h at room temperature. Then primary antibodies (see Supplementary Table 2) were diluted in 0.1 wt% BSA and incubated on sections at 4 °C overnight with 0.1 wt% BSA solution (no antibody) as negative controls. Sections were washed three times with PBS for 5 min and then incubated with corresponding secondary antibodies in 0.1 wt% BSA for 1 h at room temperature. After three washes in PBS for 5 min, sections were stained with DAPI (1:500, Themo Scientific), rinsed, and mounted with Immu-Mount™ mounting solution (Shandon). All samples were imaged using a FluoView FV1000 laser scanning confocal microscope (Olympus).

**[Ca$^{2+}$]$_i$ measurements on individual SGm**. For individual calcium measurements, SGm were manually removed from MB-hydrogel on days 7 and 14. Calcium imaging on individual clusters was performed as previously described[65,66]. Briefly, fresh AIDUCs (day 0) and SGm removed from MB-hydrogel (day 7 and day 14) were incubated with the calcium indicator 5 μM Fura-2 AM for 30 min in culture media at 37 °C. Then, the Fura-2 AM containing solution was removed and SGm were placed in imaging buffer: 137 mM NaCl, 4.7 mM KCl, 1 mM Na$_2$HPO$_4$, 1.26 mM CaCl$_2$, 0.56 mM MgCl$_2$, 5.5 mM glucose, and 10 mM HEPES (pH 7.4). Fura-2 AM-loaded SGm were placed on a 15-mm glass coverslip and mounted in a Warner chamber. These SGm were perfused with imaging buffer and alternately stimulated with 100 nM carbachol (CCh) or 100 μM ATP. SGm were alternately excited at 340 and 380 nm, and emission was captured every second at 505 nm using a Nikon microscope with a ×40 oil immersion objective and a digital camera driven by TILL Photonics software.

**[Ca$^{2+}$]$_i$ signaling assay on SGm chips**. For calcium-signaling imaging of SGm on tissue chips, calcium-binding fluorescent dye, Calbryte 520 AM was used according to the manufacturer's protocol (AATBioquest). Briefly, culture media of chips growing at day 0, 7, and 14 was replaced with 10 μM of Calbryte 520 AM ester in imaging buffer (described above) with 0.04% Pluronic F-127 and incubated for 30 min at 37 °C, 5% CO$_2$. Chips were then washed three times with buffer before fluorescent imaging. A 4 × 4 stitched BF image at ×4 magnification for each chip was taken before fluorescent imaging for ROI detection of each MB-hydrogel. Fluorescent time lapse images were captured every 5 s via a GFP light cube (Ex/Em = 490/525 nm) at 500 ms exposure time per each frame. A concentrated stock solution of CCh in imaging buffer was injected manually using a pipettor at ~60 s, achieving a final concentration of 1 μM. Time-lapse imaging was concluded after 3 min. Chips were washed 3× with buffer before imaging was performed with ATP at the final concentration of 100 μM. IPN stimulation was performed independently of CCh and ATP. A single BF image at ×2.5 magnification for each chip was taken before fluorescent imaging for ROI detection of each MB-hydrogel. Fluorescent time-lapse images were captured every 1 s via a GFP light cube (Ex/Em = 469/525 nm) at 100 ms exposure time per each frame. For IPN stimulation, a concentrated stock solution of IPN in imaging buffer was injected using a BioTek Cytation 5 Reagent Injector Module, achieving a final concentration of 10 μM. Time-lapse imaging was concluded after 3 min. Each time-lapse stack and the respective BF were processed using ImageJ. First, BF images were used to identify MB-hydrogels as ROI. BF images were processed via auto threshold, made binary with fill holes to represent each MB-hydrogel as filled circles. Secondly, the Analyze Particles tool was used to identify the circles. The ROI obtained from the previous step was utilized to only select active regions in the fluorescent time-lapse stack. Background correction was performed using the Subtract Background tool with rolling ball radius of 50 pixels. For each of the ROI, multi-measurements were conducted over time for area, mean, modal, min, and integrated fluorescent density. Both of these image processes were performed via a generated macro. Thirdly, the integrated fluorescent density over time for each MB-hydrogel was analyzed by t-test of mean between several baseline timepoints (before injection) and stimulated timepoints (after injection). Positive responsiveness of MB-hydrogels was characterized by a significant difference between the two means ($p < 0.05$). False negatives were removed when the means of stimulated timepoints were significantly lower than the baseline means.

**Analysis of human SGm amylase activity**. Human parotid SGm were incubated with 75 μg/mL of a fluorescently labeled amylase substrate (ThermoFisher, EnzChek™ Ultra Amylase Assay kit) for 30 min. After 30 min, the MB array was capped with a coverslip and the amylase substrate was removed. The fluorescence

intensity of individual MBs were measured using ImageJ with local background subtraction. The intensity of individual MBs was compared to the average intensity of the control (MB chips with substrate only) where the threshold for amylase positivity was considered as one standard deviation above the average intensity of the control and the percent of MBs positive for amylase was calculated.

**Irradiation of mice and SGm**. Mice were irradiated using a previously published methodology[138]. Briefly, mice were anesthetized with ketamine (100 mg/kg, JPH pharmaceuticals) and xylazine (10 mg/kg, Lloyd Laboratories) through intraperitoneal injection. Mice were positioned over the slit of a custom-built collimator, and received a single dose of 15 Gy from a $^{137}$Cs gamma radiation source (JL Shepard). SMG tissues were isolated 2 days after irradiation and fixed in 4% PFA at 4 °C overnight. Fixed tissues were placed in 70% ethanol at 4 °C prior to processing. Sections were then immunostained, as described above.

AIDUCs were cultured in MB-hydrogel to form SGm for 4 days and then irradiated with singular doses of 7.5, 10, and 15 Gy (JL Shepard $^{137}$Cs irradiator). For the WR1065-treated group, SGm were first pretreated with 4 mM WR1065 for 30 min prior to irradiation and for an additional 30 min after irradiation. For fractionated irradiation, an equivalent fractionated dose of 15 Gy was administered (6 Gy daily over 5 sequential days), as previously described[139]. For the group of fractionated irradiation and WR1065 treatment, SGm were pretreated with 4 mM WR1065 for 30 min prior to irradiation and for an additional 30 min after irradiation daily for 5 sequential days (see Supplementary Fig. 11 for timeline associated with fractionated irradiation). Two days post irradiation (for both singular and fractionated irradiation), cells were fixed with 4% PFA and cryosectioned (10 μm thickness) for IHC staining of DNA damage markers γH2AX and 53BP1. Staining was performed as described above except γH2AX was blocked with CAS-Block (Life Technologies) for 1 h at RT before addition of primary antibody, which was diluted with CAS-Block.

**Periodic acid-Schiff's-Alcian blue staining**. Formalin-fixed chips were rinsed with DI water, then incubated with 25 U/L diastase for 20 min at room temperature to remove glycogen. Chips were then incubated with 1% Alcian blue (Electron Microscopy Sciences, pH 2.5) for 30 min, 0.5% periodic acid (Sigma Aldrich) for 2 min, and Schiff's reagent (Sigma Aldrich) for 15 min. Rinses with DI water were performed between each step. A final rinse was performed with tap water before imaging with a Nikon Eclipse E800 equipped with a Spot Insight 12 MP CMOS camera.

**Statistics and reproducibility**. For this study, recommendations for performing rigorous experiments in biomedical sciences were observed[140]. These include redundancy in experimental design, sound statistical analysis, recognition of error, avoidance of logical traps, and intellectual honesty. All results are presented as mean ± standard error of mean and were analyzed as detailed in figure legends using SPSS or GraphPad Prism. Based on statistical methods, sample sizes were tested to ensure adequate power with $\alpha = 0.05$, $\beta = 0.2$. Differences were considered significant with $p$ values detailed in figure legends.

**Reporting summary**. Further information on research design is available in the Nature Research Reporting Summary linked to this article.

## Data availability
Data are available in the main text, supplementary materials, or from the corresponding author upon request. Source data for the graphs and charts in the figures is available in Supplementary Data 1.

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

## Acknowledgements

The authors gratefully acknowledge the support of the National Institute of Dental and Craniofacial Research (NIDCR), and the National Center for Advancing Translational Sciences (NCATS) of the National Institutes of Health, under award numbers UH3 DE027695 and UG DE027695 to D.S.W.B., C.E.O., and L.A.D., T32 ES007026 to J.A.M., F31 DE029658 to L.P., and the Training Program in Oral Sciences R90 DE022529 to H. U. and T90 DE021985 to M.H.I. The content is solely the responsibility of the authors and does not necessarily represent the official views of the National Institutes of Health. The authors would like to thank Dr. David Yule for training and access to calcium imaging equipment.

## Author contributions

Y.S., H.U. and A.S. established methods, designed and conducted experiments, analyzed data, and wrote the manuscript. L.P., J.A.M. and M.H.I. established methods, designed experiments, and reviewed the manuscript. J.R. analyzed data and reviewed the manuscript. S.D.N. coordinated with the human tissue and reviewed the manuscript. L.A.D. and C.E.O. and D.S.W.B. led the project and designed the study, and wrote the manuscript.

## Competing interests

The authors declare no competing interests.
