## [Peer Review File · Communications Biology]

Reviewers' comments:

Reviewer #1 (Remarks to the Author):

The authors build on their previous work using complex PEG hydrogels for support of primary salivary acinar cells in culture, and extend these studies to include a novel microbubble (MB) chip array for in vitro salivary tissue screening applications. Similar to other reports, the improved cell isolation procedure that preserves intact cell clusters allowed for better retention of acinar cell phenotype in vitro, including some secretory proteins, and agonist-responsive calcium dynamics, and so they call the cultured cells a salivary gland mimetic (SGm). The MB array showed enhanced ability to preserve the acinar cell phenotype using the new cell isolation procedure relative to past studies prior to the improved cell isolation procedure and is compatible with the proposed screening function of the arrays, including longitudinal studies over time. The authors extend their mouse cell findings to include primary human primary parotid cells, which exhibited similar SGm properties. To demonstrate the potential of the SGm for screening, they demonstrate that the SGm exhibit DNA damage responses to in vitro irradiation that are modulated by the radioprotector, WR-1065. The experiments are well-designed and presented and the data are convincing. The SGm have the potential to be a useful tool for the salivary gland community and for pharmaceutical companies to use for drug screening.

There are three substantial issues that reduce the potential of the SGm for drug screening, as presented here.

- 1) The acinar phenotype in the SGm is very minimal (which the authors clearly acknowledge). The acinar phenotype is substantially decreased over time up to 2 orders of magnitude lower, depending on the marker tracked.
- 2) The authors present irradiation-induced DNA damage responses that can be modulated with radioprotectors as evidence that the SGm chips are useful for drug screening. Most cells in culture exhibit irradiation-induced DNA damage responses that can be modulated with radioprotectors. There is no data provided that demonstrates the importance of the SGm technology to achieving this response.
- 3) It's not clear how difficult it is to produce the microbubble arrays and if they could be widely mass produced and used by others. The information provided is insufficient to reproduce them.

Minor:

The upregulation of Lyz2 is speculated to be a result of hypoxia; ER stress is perhaps at least as likely, which could be mentioned.

Receptor mediated calcium signaling shows an intact signaling pathway but does not show functional secretory cells, and so the conclusion is over-stated.

Many of the potential experimental design improvements proposed in the Discussion would greatly enhance the impact of this work, if they were implemented successfully.

The method of 4-hydroxy tamoxifen is not provided in the methods. Nor is the total amount provided per kg mouse weight.

No information on the average age of patient salivary gland tissue is provided.

Reviewer #2 (Remarks to the Author):

1. Brief summary of the manuscript

This work by Prof. Benoit and colleagues is a major development from their previous work using salispheres (floating SG primary cells) encapsulated into hydrogels. More recently, this group has showed that encapsulation within MMP-degradable PEG, which are degradable via specific processes, promoted epithelial polarization, lumen formation and the expression of acinar-specific markers but not Mist1 (or bhlh15a), which decreases >90%. Mist1 is required to specify the acinar phenotype in the exocrine pancreas. In the same work, the post-encapsulated acinar cells also transitioned to a duct-like phenotype and thus the acinar compartment was very limited. Thus, this submitted work tries to overcome these limitations of salispheres and primary cultures in particular the in vitro maintenance of secretory acinar cells. This would be relevant to improve functional outcomes and perform high-throughput studies to screen for novel drugs in vitro. Based on authors' previous works, a supportive microenvironment provided by MMP PEG hydrogels was utilized with microbubble (MB) array technology to assemble a modular salivary gland tissue chip platform. Microbubble (MB) arrays are not novel and have been used for the last 10 years, mainly to screen new cancer drugs in immortalized skin epithelial tissues (melanoma, SCC) in vitro. So, the application of MB for developing novel 3D screening platforms with exocrine gland primary cells, particularly from the SG is rather challenging but novel. The spherical morphology of the MB seemed to shape tissue geometry similar to functional acinus unit and maintained a niche through long-term culture to support tissue viability and function for human primary SG cells for up to 7 days (at 14 days, muscarinic receptors are dramatically reduced, approx. >90%). MBs also allowed them to fabricate high density arrays (> 4000 MB/cm²) for high-throughput and high-content screening. As mentioned by the authors, this strategy approach does not rely on any particular stem cell, but on fully differentiated salivary gland cell types. The developed platform evaluated the radiosensitivity of SG mimetics (SGm) as well as the alleviation of radiation damage (single dose only; not as per standard treatment) after using a radioprotective compound for the SG. In conclusion, the authors claim that their in vitro platform for functional SG tissue mimetics will enable high-throughput and high-content testing for new drugs, as well as mechanistic studies in radiation biology in the context of the SG.

2. Overall impression of the work

2.1 In my opinion, authors made quite a major jump in scientific rationale when deciding to use MBs together with their MMP PEG hydrogels to better maintain the acinar cell phenotype through long-term culture. They did present with literature to support the use of MBs, however these have never been used with any particular cultures of exocrine gland primary cells. The authors have presented quite an extensive literature to support the use of MBs but mainly in immortalized skin epithelial cells, where the immortal genetic background is a plus to support long-term cultures. There are several studies supporting the use of neurotrophic factors (Neurturin, GDNF) with or without ECM molecules (laminin 111 or their peptide relatives) to maintain the acinar epithelial cell compartment (PMID: 20929848; PMID: 23974175; PMID: 28784438; PMID: 29560384; PMID: 9786856, PMID: 27151393; PMID: 29095857; PMID: 28208029; PMID: 26285810; PMID: 31200143; PMID: 31028910; PMID: 31988042). Therefore, I am not sure why the MMP-degradable PEG constructs were not combined with chemically defined media with specific neurotrophic factors and/or ECM molecules (Laminin 1 peptides for example).

2.2. In terms of methodology, statistics and results, this study is scientifically sound. Though I am not sure why the innervation was not assessed particularly before and after stimulation with carbachol (a acetylcholine analog) since this cholinergic drug can stimulate neuronal sprouting (PMID: 20929848; PMID: 25805134; PMID: 24371813). Did authors assessed the neuronal population in their primary

cell cultures?

2.3. The functional activity of the SGm chips is quite limited at 14 days, since the expression of M3 muscarinic receptor falls close to zero after 14 days, as well as a major decrease in the calcium ion flow.

3. Specific comments, with recommendations for addressing each comment

3.1. Abstract: Seems somewhat confusing mixing epithelial acinar maintenance and drug development and does not really state the ultimate objectives of the study. First 2 sentences do not connect well. Please better clarify in the abstract the use of the 2 bioengineering concepts (MMP PEG and MBs).

3.2. Introduction:

3.2.1 Is there any literature supporting the use of MBs in primary cell cultures deriving from exocrine gland tissues? Please provide evidence.

3.2.2 There are very few cited works on SG organoids/salispheres since these can also be used as in vitro screening platforms in the very near future (PMID: 20929848; PMID: 25805134; PMID: 24371813; PMID: 26887347)

3.3. Methods have an overall satisfactory scientific quality, however I do have these 2 major concerns:

3.3.1 There was no scientific rationale given to the fact that no adrenergic agonist (e.g. isoproterenol) was used to evaluate for sympathetic stimulation of the SGm. This has been done before for 3D culture SG platforms and organoids (PMID: 30025245; PMID: 26285810; PMID: 30666813)

3.3.2 There was no scientific rationale for the use of single dose of radiation for instead of the standard fractionated dose regimen for head and neck cancers?

3.4. Results: ATP stimulation as a purinergic agonist: Extracellular purine nucleotides and nucleosides have biological effects in a variety of cell and tissue types including adult and fetal keratinocytes, fibroblasts, melanocytes, mast cells, Langerhans cells, and Meissner's corpuscles, hair follicles, sweat glands, smooth muscle and endothelial cells of skin vessels, etc. Therefore, how specific is this stimulation for the SG epithelia which may have ADRA2 receptors... would it rather be better to use an adrenergic agonist to determine if it can respond to adrenergic agonists like isoproterenol? This has been done in other recent publications: Reference 45 was used to validate the use of purinergic agonist like ATP; however, this was a P2X(7) KO mouse study that despite showing an interaction between purinergic and cholinergic receptor signaling, it also shows that not all exocrine glands are affected and function of this receptor is suppressed in female salivary glands. We know that sexual dimorphism in the SG occurs in rodents but not in humans. How can we translate these results to your human chips?

Moreover, in Fig 4b, the expression of P2x7 falls dramatically (more than 50% after 5 days and has a very slight recovery) - what do you think it's occurring? In Fig 4c, P2y2 expression also decreases after 7 days; and in Fig 4a, the expression of M3 muscarinic receptor falls close to zero after 14 days. Please explain the impact of these results in the ultimate applications of the final chip if one wants to test their functional activity upon neural stimulation. Figs 4e-4g gives a better idea about the CCh response (more specific to SG acinar cells than ATP as discussed earlier), which seems to decrease substantially over time (from 0 to 14 days) as per the Y axis scale.

3.5. In the introduction, authors state "However, function is only maintained for 24 - 48 h, which is not amenable to evaluate secretory dysfunction in general, and particularly, as a consequence of radiation damage, which requires longer times to realize physiological dysfunction". In view of my

concerns on item 3.4 and this statement by the authors, if the expression of M3 receptors is close to zero (Fig 4a) and the SGM chips functional activity upon stimulation is substantially reduce after 14 days, do you think you still can assess function after a regimen of fractionated radiotherapy with 21 days in duration (standard of patient care for head and neck cancers (HNC). Please look at current standards of care before and during covid-19 times:

<https://www.sciencedirect.com/science/article/pii/S0360301620310348>

You have used single dose of radiation (15Gy) for screening a radioprotective drug. Would the results vary dramatically if one used a fractionated dose regimen (6 Gy for 5 days) as per PMID: 29560384 ?

3.5. Discussion:

3.5.1. The potential use of chemically defined media with neurotrophic factors (Neurturin, GDNF) should be discussed to optimize the acinar cell maintenance in their platform. Authors did briefly cite reference 19 (by Vining et al. 2019), which supported the use of Neurturin and Laminin 111 gel, but did not discuss the true impact of such approach if used in their platform. Also, previous studies supporting the use of Neurturin should be cited as they are several (PMID: 29560384; PMID: 20929848; PMID: 23974175 as well as for GDNF (PMID: 28784438; PMID: 25036711).

3.5.2. This statement on page 16 "While the media used in this study included FGF2, the addition of FGF7, FGF10 or other combinations of FGFs could further enhance tissue function." is quite relevant based on the FGF7 and FGF10 KO mice (have SG hypoplasia or aplasia), however no citations were added to this important and perhaps future strategy being pursued by the authors. For example, the work by Adine et al (PMID: 30025245) supports the use of FGF10 in the development of epithelial SG-like organoids.

3.5.3. In addition, the sentence that follows (on page 16) "Several studies have also documented the effects of neural factors in promoting the branching and development of glandular tissues, although these factors may not be sufficient (ref 108-110)" cites mainly references for mouse fetal or developmental models which have little relationship to the authors primary cell cultures. Therefore, why not adding neural factors to their human parotid SGM chips?

3.5.4. As per my last 2 concerns, the discussion lacks a more in-depth analysis of the current literature as recent papers do support the use of "neural factors", FGF7/10 as well as specific ECM molecules (Laminin 111 for e.g.).

3.5.5. Please discuss your option for single dose instead of standard fractionated dose regimen when testing the WR-1065 radioprotective agent. How can the secretory agonist-mediated Ca²⁺-flux be maintained if standard radiotherapy regimens go well over 14 days?

3.5.6. Why would it be advantageous to have intercalated ducts in the chips and not other duct types that have a larger stem cell niche (like excretory ducts shown in PMID: 26378247) ?

Reviewer #3 (Remarks to the Author):

Song et al Nature Biology Communications Review

Comments:

Song et al present an interesting manuscript addressing a problem central to salivary gland (SG) research in the context of radiation-induced damage, namely that saliva producing cells, the acinar cells, are prone to loss of phenotype in in vitro culture situations. This interferes with our ability as a radiation research community to screen potential new drug targets, and ameliorate radiation-induced hyposalivation. The manuscript is well written. I have some major and minor comments regarding the manuscript, that I would like to see addressed before publication in Nature Biology Communications.

Major:

1. Whilst I appreciate that the data of Song et al has significantly improved the culture conditions necessary to maintain SG acinar cell phenotype in vitro, the nature of the paper message, i.e. that acinar cells lose their phenotype in vitro, is not per se novel. The authors have acknowledged this themselves. I question whether this is novel enough for Nature Biology Communications.

2. Continuing on this theme, the abstract does not in my opinion truly reflect the data presented. Namely that with all markers and experiments examined, acinar cells still lose their phenotype in vitro. The abstract suggests that acinar phenotype is maintained exactly as that of the native tissue, something I would like to be addressed.

3. There is some confusion with reference in ductal cell types in the manuscript. Namely:

i) Are the authors claiming that they do not seed striated duct-type structures, which they state are present in their isolated cell types, into the MBs? Are they excluding these structures somehow from their analysis?

ii) In the mouse system, the authors use K5 and K7 interchangeably to denote 'ductal' cells (page 6, end of second paragraph, p7 paragraph 1). Please specify precisely to which ductal cell subpopulation you refer here. K5 is commonly used for example to denote luminal cells of the striated ducts, and K7 for IDs cells and luminal striated duct cells. If the authors wish to use both markers, in the case of Fig. 2m, please perform then qPCR and staining for both K5 and k7.

iii) The authors state that K5 expression increases in time in the MB system. Again, with reference to the type of ductal cell you are referring to, what is the inference from this statement? Which type of ductal cells are increasing in prevalence? (Page 6).

4. What is the authors hypothesis about the Mist1 expression? They state that Mist1 expression increases in MB culture (Day 0 24% to Day 4/6 40 %), whilst acinar cell functional markers decrease. Based on the previous work of the Ovitt group, are the authors insinuating that a acinar cell progenitor type population results from their culture system, at least partially?

Minor comments:

1. Use of LIVE-DEAD stain: I don't agree with the method of quantification. '>90% of MBs within each array contain viable cells' – this could be anything from 1% to 100% viable cells. Is there any way you can quantify the percentage of viable cells in the MBs? Find this data a little misleading otherwise.

2. Fig3e-f: Amylase staining at D14 looks really bright, although authors state that it is less than Day 0. Could a Day 0 panel be added to this figure for comparison?

3. Gross morphology of the acinar cells in MB chips still not extremely convincing, although markers are expressed. I miss the characteristic triangular acinar cell shape. Have you tried doing an H&E

staining on the MB?

4. P8 paragraph 2: 'P2Y2 expression maintained consistent expression through day 14' – I find this a very strong statement, based on the variability of data in Figure 4c. Can you tone this down?

5. Fig4K: can the authors make the fluorescence panels larger – hard to make out the change in colour from such small images.

6. Fig. 5n – could the authors show the raw data of the human amylase secretion work, if possible?

COMMSBIO-20-1718-T

We would like to thank the reviewers for their thorough and thoughtful reviews, which have resulted in a much-improved manuscript. Please find below our point-by-point responses in red to address concerns that were raised including reference to changes made within the revised manuscript.

Reviewer #1 (Remarks to the Author):

The authors build on their previous work using complex PEG hydrogels for support of primary salivary acinar cells in culture, and extend these studies to include a novel microbubble (MB) chip array for in vitro salivary tissue screening applications. Similar to other reports, the improved cell isolation procedure that preserves intact cell clusters allowed for better retention of acinar cell phenotype in vitro, including some secretory proteins, and agonist-responsive calcium dynamics, and so they call the cultured cells a salivary gland mimetic (SGm). The MB array showed enhanced ability to preserve the acinar cell phenotype using the new cell isolation procedure relative to past studies prior to the improved cell isolation procedure and is compatible with the proposed screening function of the arrays, including longitudinal studies over time. The authors extend their mouse cell findings to include primary human primary parotid cells, which exhibited similar SGm properties. To demonstrate the potential of the SGm for screening, they demonstrate that the SGm exhibit DNA damage responses to in vitro irradiation that are modulated by the radioprotector, WR-1065. The experiments are well-designed and presented and the data are convincing. The SGm have the potential to be a useful tool for the salivary gland community and for pharmaceutical companies to use for drug screening.

There are three substantial issues that reduce the potential of the SGm for drug screening, as presented here.

1) The acinar phenotype in the SGm is very minimal (which the authors clearly acknowledge). The acinar phenotype is substantially decreased over time up to 2 orders of magnitude lower, depending on the marker tracked.

As acknowledged in the manuscript, long-term maintenance of the acinar phenotype is still a challenge despite our findings. Development of hydrogel-based platforms to culture salivary gland cells has showed great potential to maintain acinar phenotype. Our and others' previous studies have shown that 3D cultured salivary gland cells in hydrogels (including Matrigel, hyaluronan-based hydrogels, decellularized extracellular matrix hydrogels, fibrin hydrogels, and poly(ethylene glycol) based hydrogel) showed improved maintenance of acinar marker expression, including α -amylase, Aqp5, Mist1, and Nkcc1 compared with traditional 2D culture [3-11]. Nevertheless, significant reductions in acinar marker expression of Mist1, amylase, and Aqp5 are still observed [1, 2]. In the SGm, an 8-fold and 3-fold increase of Mist1 expression level is observed at days 7 and 14, respectively, compared to our previous studies, which utilized macroscale MMP-degradable hydrogels and pre-culture of cells to form salispheres [11]. These data indicate the advantage of SGm in maintaining acinar phenotype. Despite this advantage, there is still reduction in acinar gene expression. However, data show IHC staining of proteins consistent with the acinar phenotype, such as AQP5 and amylase, as well as CCh, ATP, and IPN stimulated calcium signaling, indicating successful maintenance of secretory function. Nevertheless, we recognize that there is still much work to be done to further develop this and other strategies to enable long-term function of tissue ex vivo. A more comprehensive discussion summarizing these points, as well as ideas to continue to improve function, are included in the revised discussion.

2) The authors present irradiation-induced DNA damage responses that can be modulated with radioprotectors as evidence that the SGm chips are useful for drug screening. Most cells in culture exhibit irradiation-induced DNA damage responses that can be modulated with radioprotectors. There is no data provided that demonstrates the importance of the SGm technology to achieving this response.

We agree that the radioprotective response showing lower numbers of gamma H2AX is not specific to salivary gland cells. However, amifostine is used clinically as a radioprotectant for the salivary glands, so

our goal was to establish that the SGm respond similar to whole gland tissue. This was considered critical, as the SGm are primary cells and not established cell cultures. In combination with the data showing that SGm mimic other salivary gland traits, amifostine protection from the DNA damage response supports the approach of using primary cells for drug screening. Further work is ongoing to determine additional markers that may be measured to assess protection of functional activity.

3) It's not clear how difficult it is to produce the microbubble arrays and if they could be widely mass produced and used by others. The information provided is insufficient to reproduce them.

Fabrication of microbubble technology has been published and relevant articles have been referenced within the manuscript (see Results, middle of page 5 and refs. 41-42). We also have added a figure in the supplemental section (Fig S3) that illustrates the basic fabrication process. Microbubble chips are commercially available through Nidus MB Technologies and we have added this information to the relevant figure captions in the revised manuscript.

Giang UB, Lee D, King MR, DeLouise LA. Microfabrication of cavities in polydimethylsiloxane using DRIE silicon molds. *Lab Chip*. 2007 Dec;7(12):1660-2. doi: 10.1039/b714742b. Epub 2007 Oct 12. PMID: 18030383; PMCID: PMC2587163.

Giang UB, Jones MC, Kaule MJ, Virgile CR, Pu Q, Delouise LA. Quantitative analysis of spherical microbubble cavity array formation in thermally cured polydimethylsiloxane for use in cell sorting applications. *Biomed Microdevices*. 2014 Feb;16(1):55-67. doi: 10.1007/s10544-013-9805-5. PMID: 24037662.

Minor:

The upregulation of Lyz2 is speculated to be a result of hypoxia; ER stress is perhaps at least as likely, which could be mentioned.

We appreciate the reviewer pointing out this alternative interpretation of our data. We have updated our discussion regarding Lyz2 upregulation in line with the potential of ER stress underpinning its upregulation (see 1st paragraph of p. 8).

Receptor mediated calcium signaling shows an intact signaling pathway but does not show functional secretory cells, and so the conclusion is over-stated.

We agree with the reviewer that intact signaling does not necessarily prove that the cells are secretory. We have reworded the sentence to highlight the intact pathway, which suggests functional acinar cells (see end of p. 9).

Many of the potential experimental design improvements proposed in the Discussion would greatly enhance the impact of this work, if they were implemented successfully.

We agree that the discussion highlights design improvements to further enhance function of the SGm. Our ongoing experiments are indeed focused on many of these improvements, which will be reported as such. Nevertheless, the studies are beyond the scope of this initial report, which is focused on proof of concept of SGm development and use.

The method of 4-hydroxy tamoxifen is not provided in the methods. Nor is the total amount provided per kg mouse weight.

We apologize for omissions in the experimental approach of this study. The 4-hydroxytamoxifen was added in vitro to induce the activation of Mist1-Cre and expression of RFP in SGMs. Due to the significantly reduced levels of Mist1 in encapsulated SGMs, these experiments were done to demonstrate

that there are cells remaining that still have active expression of Mist1, as measured by the activity of the Cre which is driven by the Mist1 promoter.

No information on the average age of patient salivary gland tissue is provided.

This information has been added to the manuscript (see Methods, middle of p. 18).

Reviewer #2 (Remarks to the Author):

1. Brief summary of the manuscript

This work by Prof. Benoit and colleagues is a major development from their previous work using salispheres (floating SG primary cells) encapsulated into hydrogels. More recently, this group has showed that encapsulation within MMP-degradable PEG, which are degradable via specific processes, promoted epithelial polarization, lumen formation and the expression of acinar-specific markers but not Mist1 (or bhlh15a), which decreases >90%. Mist1 is required to specify the acinar phenotype in the exocrine pancreas. In the same work, the post-encapsulated acinar cells also transitioned to a duct-like phenotype and thus the acinar compartment was very limited. Thus, this submitted work tries to overcome these limitations of salispheres and primary cultures in particular the in vitro maintenance of secretory acinar cells. This would be relevant to improve functional outcomes and perform high-throughput studies to screen for novel drugs in vitro. Based on authors' previous works, a supportive microenvironment provided by MMP PEG hydrogels was utilized with microbubble (MB) array technology to assemble a modular salivary gland tissue chip platform. Microbubble (MB) arrays are not novel and have been used for the last 10 years, mainly to screen new cancer drugs in immortalized skin epithelial tissues (melanoma, SCC) in vitro. So, the application of MB for developing novel 3D screening platforms with exocrine gland primary cells, particularly from the SG is rather challenging but novel. The spherical morphology of the MB seemed to shape tissue geometry similar to functional acinus unit and maintained a niche through long-term culture to support tissue viability and function for human primary SG cells for up to 7 days (at 14 days, muscarinic receptors are dramatically reduced, approx. >90%). MBs also allowed them to fabricate high density arrays (> 4000 MB/cm²) for high-throughput and high-content screening. As mentioned by the authors, this strategy approach does not rely on any particular stem cell, but on fully differentiated salivary gland cell types. The developed platform evaluated the radiosensitivity of SG mimetics (SGm) as well as the alleviation of radiation damage (single dose only; not as per standard treatment) after using a radioprotective compound for the SG. In conclusion, the authors claim that their in vitro platform for functional SG tissue mimetics will enable high-throughput and high-content testing for new drugs, as well as mechanistic studies in radiation biology in the context of the SG.

2. Overall impression of the work

2.1 In my opinion, authors made quite a major jump in scientific rationale when deciding to use MBs together with their MMP PEG hydrogels to better maintain the acinar cell phenotype through long-term culture. They did present with literature to support the use of MBs, however these have never been used with any particular cultures of exocrine gland primary cells. The authors have presented quite an extensive literature to support the use of MBs but mainly in immortalized skin epithelial cells, where the immortal genetic background is a plus to support long-term cultures. There are several studies supporting the use of neurotrophic factors (Neurturin, GDNF) with or without ECM molecules (laminin 111 or their peptide relatives) to maintain the acinar epithelial cell compartment (PMID: 20929848; PMID: 23974175; PMID: 28784438; PMID: 29560384; PMID: 9786856, PMID: 27151393; PMID: 29095857; PMID: 28208029; PMID: 26285810; PMID: 31200143; PMID: 31028910; PMID: 31988042). Therefore, I am not sure why the MMP-degradable PEG constructs were not combined with chemically defined media with specific neurotrophic factors and/or ECM molecules (Laminin 1 peptides for example).

We agree with the reviewer that addition of neurotrophic factors and other matrix cues would be beneficial to support long-term function of the SGm, as detailed in the discussion and rationalized by the reviewers

suggested references (see addition of references and insight from these studies found now within the discussion). Indeed, our ongoing experiments are focused on many of these improvements, which will be reported as such. Nevertheless, the studies are beyond the scope of this initial report, which is focused on proof of concept of SGm development and use.

2.2. In terms of methodology, statistics and results, this study is scientifically sound. Though I am not sure why the innervation was not assessed particularly before and after stimulation with carbachol (a acetylcholine analog) since this cholinergic drug can stimulate neuronal sprouting (PMID: 20929848; PMID: 25805134; PMID: 24371813). Did authors assessed the neuronal population in their primary cell cultures?

We assessed isolated AIDUCs for expression of the neuronal marker β III-tubulin. However, threshold cycles were very high (>37) and for 4/6 samples analyzed, undetectable (see Figure S4). From this data, we conclude that there is most likely not a neuronal population that is seeded to form the SGms (see page 6 for inclusive discussion of this new data).

2.3. The functional activity of the SGm chips is quite limited at 14 days, since the expression of M3 muscarinic receptor falls close to zero after 14 days, as well as a major decrease in the calcium ion flow.

We agree and have highlighted in the manuscript that function of the SGm decreases over the time course of our study (14 days). However, in our proof of principle data reported herein, 14 days enables testing of radioprotective efficacy of the active metabolite of amifostine (WR-1065). Therefore, longer studies, which should be enabled by improvements in the overall platform using soluble and matrix cues, are not necessary to perform drug screening in the SGm functional time frame.

3. Specific comments, with recommendations for addressing each comment

3.1. Abstract: Seems somewhat confusing mixing epithelial acinar maintenance and drug development and does not really state the ultimate objectives of the study. First 2 sentences do not connect well. Please better clarify in the abstract the use of the 2 bioengineering concepts (MMP PEG and MBs).

We thank the reviewer for this critique and have revised the abstract to improve the flow and linkage of concepts.

3.2. Introduction:

3.2.1 Is there any literature supporting the use of MBs in primary cell cultures deriving from exocrine gland tissues? Please provide evidence.

Indeed, the reviewer points out an exciting innovation of our current study in that this is the first instance whereby MB culture is utilized for exocrine tissues. So, the application of MB for developing novel 3D screening platforms with exocrine gland primary cells, particularly from the SG is rather challenging but novel. We have added this point to the introduction (top of p. 4).

3.2.2 There are very few cited works on SG organoids/salispheres since these can also be used as in vitro screening platforms in the very near future (PMID: 20929848; PMID: 25805134; PMID: 24371813; PMID: 26887347)

We agree that SG spheres may also be used for in vitro screening platforms and have added this to the manuscript (see revised introduction, bottom of p. 2). However, our approach for this study was based on the difficulty with characterization and imaging of individual spheres grown in cultures. In previously published work, we showed that SG spheres can be formed and cultured in hydrogels (Shubin et al. 2015). However, we also found that although dissociated cells efficiently formed spheres (Varghese et al. 2019), they did not maintain a high percentage of acinar cells over time, and in addition, they exhibited alterations in cell phenotype. For this reason, our focus in the current study has been to maintain acini/ID structures that better mimic SG function.

3.3. Methods have an overall satisfactory scientific quality, however I do have these 2 major concerns:

3.3.1 There was no scientific rationale given to the fact that no adrenergic agonist (e.g. isoproterenol) was used to evaluate for sympathetic stimulation of the SGm. This has been done before for 3D culture SG platforms and organoids (PMID: 30025245; PMID: 26285810; PMID: 30666813)

We thank the reviewer for pointing out this important oversight in our initial report. In response, we have characterized beta-adrenergic receptor expression and explored isoproterenol stimulation in the mouse SGm. Both data sets are consistent with expression and stimulus data shown in Fig. 4. Receptor expression decreases over time but is still detectable at day 14. Additionally, isoproterenol induces a 2-fold increase in MB calcium signaling at day 7 and statistically equivalent responsive MB at day 14 normalized to day 0, consistent with adrenergic receptor activity and data reported for CCh and ATP stimuli in Fig. 4m (These data have been included in Supplemental Fig. S9).

3.3.2 There was no scientific rationale for the use of single dose of radiation for instead of the standard fractionated dose regimen for head and neck cancers?

The single dose of radiation (15 Gy) has been used in many studies as a model that produces the same endpoints as clinically fractionated doses (i.e. loss of acinar cells). We agree that fractionated doses have great significance clinically and we have performed this analysis, as detailed in the revised manuscript (see the updated Methods section and Fig. 6). Importantly, the SGm exhibits similar WR-1065 radioprotection in the context of dose-matched fractionated radiation versus singular 15 Gy doses.

3.4. Results: ATP stimulation as a purinergic agonist: Extracellular purine nucleotides and nucleosides have biological effects in a variety of cell and tissue types including adult and fetal keratinocytes, fibroblasts, melanocytes, mast cells, Langerhans cells, and Meissner's corpuscles, hair follicles, sweat glands, smooth muscle and endothelial cells of skin vessels, etc. Therefore, how specific is this stimulation for the SG epithelia which may have ADRA2 receptors... would it rather be better to use an adrenergic agonist to determine if it can respond to adrenergic agonists like isoproterenol? This has been done in other recent publications: Reference 45 was used to validate the use of purinergic agonist like ATP; however, this was a P2X(7) KO mouse study that despite showing an interaction between purinergic and cholinergic receptor signaling, it also shows that not all exocrine glands are affected and function of this receptor is suppressed in female salivary glands. We know that sexual dimorphism in the SG occurs in rodents but not in humans. How can we translate these results to your human chips?

We agree that investigation of isoproterenol stimulation of ADRA2 receptors is important to investigate for SGm functional analyses and have performed the suggested experiments. We now include data from these studies in Fig. S9, as detailed above. As the reviewer has stated, we do not have evidence that there is sexual dimorphism of the SG in humans. Human tissues were obtained from both male and female patients. We note that the response of both mouse and human SGm to ATP were similar at days 7 and 14 (see Fig. 4i,j and Fig. 5s,t), so do not think that sexual differences have a bearing on the results.

Moreover, in Fig 4b, the expression of P2x7 falls dramatically more than 50% after 5 days and has a very slight recovery) - what do you think it's occurring? In Fig 4c, P2y2 expression also decreases after 7 days; and in Fig 4a, the expression of M3 muscarinic receptor falls close to zero after 14 days. Please explain the impact of these results in the ultimate applications of the final chip if one wants to test their functional activity upon neural stimulation. Figs 4e-4g gives a better idea about the CCh response (more specific to SG acinar cells than ATP as discussed earlier), which seems to decrease substantially over time (from 0 to 14 days) as per the Y axis scale.

Indeed, P2x7 expression is reduced at early timepoints in the study. We believe this may be due to initial stress response of tissue isolation and reorganization within the SGm, which is consistent with human SGm data (Fig. 5). However, it is important to note that gene expression trends only predict downstream protein expression, which is functionally responsible for secretory behaviors. Membrane receptors,

including P2x7 and M3, typically have long half-lives, therefore ATP-stimulated calcium signaling observed in SGM is maintained over the same timeframe. Nevertheless, the culmination of data reported herein suggest that SGM function is diminishing over culture time, especially by day 14. Hence, our current efforts are focused on enhancing longitudinal tissue function by additional of soluble and matrix cues, as detailed in the updated discussion, to further enhance SGM robustness. More comprehensive discussion of the receptor expression and calcium signaling data and potential implications are included now on page 10 and within the discussion.

3.5. In the introduction, authors state "However, function is only maintained for 24 - 48 h, which is not amenable to evaluate secretory dysfunction in general, and particularly, as a consequence of radiation damage, which requires longer times to realize physiological dysfunction". In view of my concerns on item 3.4 and this statement by the authors, if the expression of M3 receptors is close to zero (Fig 4a) and the SGM chips functional activity upon stimulation is substantially reduce after 14 days, do you think you still can assess function after a regimen of fractionated radiotherapy with 21 days in duration (standard of patient care for head and neck cancers (HNC). Please look at current standards of care before and during covid-19 times: <https://www.sciencedirect.com/science/article/pii/S0360301620310348> You have used single dose of radiation (15Gy) for screening a radioprotective drug. Would the results vary dramatically if one used a fractionated dose regimen (6 Gy for 5 days) as per PMID: 29560384 ?

First, to clarify, the short-term function of 24-48 h is not describing the SGM in this study, but rather refers to freshly prepared salivary gland slices in a published study (Warner et al. Am J Physiol Gastrointest Liver Physiol, 2008). The SGM developed here show tissue function up to day 14, although function decreased by the last timepoint investigated. To address the first issue, the experimental design using the current iteration of chips would be to use a single dose of radiation within the first 7 days of radiation, and to assess radioprotection for up to 14 days. As the ability to enhance longer cell viability and gene expression profiles is increased, screening could be transitioned to using fractionated doses. To address the second issue, we have now included data to show that SGM indeed can be used to investigate radioprotection in the context of clinically relevant fractionated irradiation (see Fig. 6).

3.5. Discussion:

3.5.1. The potential use of chemically defined media with neurotrophic factors (Neurturin, GDNF) should be discussed to optimize the acinar cell maintenance in their platform. Authors did briefly cite reference 19 (by Vining et al. 2019), which supported the use of Neurturin and Laminin 111 gel, but did not discuss the true impact of such approach if used in their platform. Also, previous studies supporting the use of Neurturin should be cited as they are several (PMID: 29560384; PMID: 20929848; PMID: 23974175 as well as for GDNF (PMID: 28784438; PMID: 25036711).

We agree with the reviewer that addition of neurotrophic factors and other matrix cues would be beneficial to support long-term function of the SGM, as detailed in the discussion and augmented by the reviewers suggested references (see addition of references and insight from these studies found now within the discussion). Indeed, our ongoing experiments are focused on many of these improvements, which will be reported as such. Nevertheless, the studies are beyond the scope of this initial report, which is focused on proof of concept of SGM development and use.

3.5.2. This statement on page 16 "While the media used in this study included FGF2, the addition of FGF7, FGF10 or other combinations of FGFs could further enhance tissue function." is quite relevant based on the FGF7 and FGF10 KO mice (have SG hypoplasia or aplasia), however no citations were added to this important and perhaps future strategy being pursued by the authors. For example, the work by Adine et al (PMID: 30025245) supports the use of FGF10 in the development of epithelial SG-like organoids.

Thank you for pointing out our lack of citations. The discussion has been updated accordingly.

3.5.3. In addition, the sentence that follows (on page 16) "Several studies have also documented the effects of neural factors in promoting the branching and development of glandular tissues, although these factors may not be sufficient (ref 108-110)" cites mainly references for mouse fetal or developmental models which have little relationship to the authors primary cell cultures. Therefore, why not adding neural factors to their human parotid SGm chips?

Thank you for the suggested future directions. These experiments are underway and beyond the scope of the current report.

3.5.4. As per my last 2 concerns, the discussion lacks a more in-depth analysis of the current literature as recent papers do support the use of "neural factors", FGF7/10 as well as specific ECM molecules (Laminin 111 for e.g.).

We apologize that our discussion lacked this in-depth analysis. We have added additional references and discussion but these comments are concise due to page constraints.

3.5.5. Please discuss your option for single dose instead of standard fractionated dose regimen when testing the WR-1065 radioprotective agent. How can the secretory agonist-mediated Ca²⁺-flux be maintained if standard radiotherapy regimens go well over 14 days?

We agree with the reviewer and have provided additional data within the revised manuscript (Fig. 6) that shows similar WR-1065 protection against fractionated doses compared with single 15 Gy doses. Given the chosen fractionated model is bioequivalent to 15 Gy, which recapitulates human disease sequelae, we are confident that drug screening using the SGm will elucidate promising radioprotective compound hits.

3.5.6. Why would it be advantageous to have intercalated ducts in the chips and not other duct types that have a larger stem cell niche (like excretory ducts shown in PMID: 26378247) ?

These MB/tissue chips are dependent on the maintenance of isolated cells, not on the generation of new cells from stem cell activity. Acinar-intercalated duct complexes have been established as an ideal in vitro model for studying secretory responses of salivary gland, but they were found to remain functional for only hours after isolation [Quissell et al. 1986]. As in the aforementioned study, our intent was to leave the acini intact and connected to the intercalated ducts to maintain the acinar cell polarity and secretory function, while the goal of this study was to extend the functional timeframe. When cultured under 2D or 3D conditions, isolated salivary gland cells adopt ductal phenotypes, such as expression of K5 and K14, and become the predominant cell type in the cultures [e.g. Srinivasan et al. 2016]. For this reason, we have not attempted to isolate only ducts for SGm development.

Reviewer #3 (Remarks to the Author):

Song et al Nature Biology Communications Review

Comments:

Song et al present an interesting manuscript addressing a problem central to salivary gland (SG) research in the context of radiation-induced damage, namely that saliva producing cells, the acinar cells, are prone to loss of phenotype in in vitro culture situations. This interferes with our ability as a radiation research community to screen potential new drug targets, and ameliorate radiation-induced hyposalivation. The manuscript is well written. I have some major and minor comments regarding the manuscript, that I would like to see addressed before publication in Nature Biology Communications.

Major:

1. Whilst I appreciate that the data of Song et al has significantly improved the culture conditions necessary to maintain SG acinar cell phenotype in vitro, the nature of the paper message, i.e. that acinar

cells lose their phenotype in vitro, is not per se novel. The authors have acknowledged this themselves. I question whether this is novel enough for Nature Biology Communications.

We acknowledge that the central message of this manuscript was not clearly recognized in our initial submission. We have revised the manuscript to stress the two major advances in this work: (1) When intracellular contacts are maintained during isolation and cultured under the conditions described, the acinar cells maintain secretory phenotype longer than in previous studies; (2) arrays of these cells cultured in MBs allow us to characterize phenotypic and functional parameters in a high-throughput manner, as well as to employ them for drug testing. This level of screening has not previously been demonstrated for salivary gland tissue.

2. Continuing on this theme, the abstract does not in my opinion truly reflect the data presented. Namely that with all markers and experiments examined, acinar cells still lose their phenotype in vitro. The abstract suggests that acinar phenotype is maintained exactly as that of the native tissue, something I would like to be addressed.

We appreciate the reviewer's critique and have revised the abstract to state both the goal and the outcomes of this project more clearly.

3. There is some confusion with reference in ductal cell types in the manuscript. Namely:

i) Are the authors claiming that they do not seed striated duct-type structures, which they state are present in their isolated cell types, into the MBs? Are they excluding these structures somehow from their analysis?

We agree with the reviewer that the approach used to isolate cells for SGm development was unclear. The cell clusters seeded into the MB arrays included all cell types that were present after the isolation and size selection were done. This most likely included striated duct structures, such as that seen in Figure S2c, which is stained with the luminal duct cell marker, K7. The only ducts which might be excluded could be the largest excretory ducts, if they are not sufficiently dissociated. However, no effort was made to purposefully exclude them. The first paragraph of the results has been revised accordingly to ensure clarity of this point.

ii) In the mouse system, the authors use K5 and K7 interchangeably to denote 'ductal' cells (page 6, end of second paragraph, p7 paragraph 1). Please specify precisely to which ductal cell subpopulation you refer here. K5 is commonly used for example to denote luminal cells of the striated ducts, and K7 for ID cells and luminal striated duct cells. If the authors wish to use both markers, in the case of Fig. 2m, please perform then qPCR and staining for both K5 and k7.

We acknowledge the interchangeable use of K5 and K7 in the initial submission. We have clarified the use of both markers in the resubmission and have also included immunostaining and qPCR for both K5 and K7. Please find the added data in Fig. 2 and associated discussion in the middle of p. 7. These data show inconsistencies between K7 and K5 expression and immunostaining, whereby expression levels reach 6- and 12-fold day 0 values, yet immunostaining suggests only modest positivity within the SGm. This can be explained by cell outgrowth from the MB observed beyond day 7 of a cytokeratin-expressing population that resides on top of the chip (see Supp Fig. 8). RNA is extracted from the entire chip rather than only the SGm, therefore, the inconsistency may be due to the outgrowth cells overwhelming gene expression trends for K5 and K7.

iii) The authors state that K5 expression increases in time in the MB system. Again, with reference to the type of ductal cell you are referring to, what is the inference from this statement? Which type of ductal cells are increasing in prevalence? (Page 6).

It has been established that K5 and K7 (new data Fig. 2) expression increases with culture of salivary gland cells in vitro, as well as in injured or irradiated glands [Watanabe et al. 2017; Sullivan et al. 2005],

and disease [Pringle et al 2020]. We have previously reported this in hydrogel-cultured spheres [Shubin et al. 2019]. It is most likely that cells activate K5 and K7 expression under stress [Pan et al. 2013 PMID: 23270662; Toivola et al. 2015 PMID: 25599598], which may result in outgrowth, as observed in Supp. Fig. 8.

4. What is the authors hypothesis about the Mist1 expression? They state that Mist1 expression increases in MB culture (Day 0 24% to Day 4/6 40 %), whilst acinar cell functional markers decrease. Based on the previous work of the Ovitt group, are the authors insinuating that a acinar cell progenitor type population results from their culture system, at least partially?

We apologize that these results were confusing with regard to the overall conclusions of the paper, and have removed the quantification for these experiments from the manuscript. The results show that at both day 4 and 6 there were some cells that expressed Cre and responded to tamoxifen. This indicates that the SGM do maintain some Mist1-positive cells. However, the values reflected the number of MBs that contained RFP positive cells on each day tested, and the data were obtained from separate isolations. They should therefore not have been directly compared and should not be interpreted as measures of Mist1 expression levels. Unfortunately, due to COVID-19 related lab shutdowns and restrictions, these experiments cannot be replicated in a reasonable timeframe. As the findings are not crucial to the overall study conclusions, we have removed this data from the resubmission.

Minor

1. Use of LIVE-DEAD stain: I don't agree with the method of quantification. '>90% of MBs within each array contain viable cells' – this could be anything from 1% to 100% viable cells. Is there are way you can quantify the percentage of viable cells in the MBs? Find this data a little misleading otherwise.

We agree that the LIVE/DEAD quantification is unconventional. However, due to multicellular 3D aggregates that form within the chips, robust LIVE/DEAD quantification by enumeration of individual cells is challenging and fraught with inconsistency. This is due to non-discrete cells, signal bleed through by different levels of tissue, and constraints with imaging two different channels with different integration times, whereby neither intensity level nor excitation efficiency is identical. Therefore, we elected to employ the alternative yes/no on a per MB basis, which simplifies quantification and alleviates these inconsistencies. However, it is clear from the representative images (see Fig. 1 j-k, Fig. 5a-c) that the majority of cells of the imaged tissues in individual MBs are alive.

2. Fig3e-f: Amylase staining at D14 looks really bright, although authors state that is it less than Day 0. Could a Day 0 panel be added to this figure for comparison?

Thank you for suggesting this important control for comparison. The Day 0 Amylase control is now included in Fig. 3 along with Day 0 Pip controls.

3. Gross morphology of the acinar cells in MB chips still not extremely convincing, although markers are expressed. I miss the characteristic triangular acinar cell shape. Have you tried doing an H&E staining on the MB?

The characteristic triangular shape of acinar cells is quite prominent in AIDUCs shown in Supplemental Figures 2a, b, and d. However, as has been shown by many groups, the morphology of acinar cells is not well maintained in vitro under various culture conditions [see Srinivasan 2016 PMID: 28170182; Pringle et al. 2016 PMID: 26887347; Adine et al. 2018; Seo,Tran 2019; Nam et al. 2019 PMID: 31285850; Shubin et al. 2017 PMID:28039063], although these reports showed maintenance of acinar cell-specific gene expression, suggesting that acinar cells are present.

We have also performed Periodic acid-Schiff's – Alcian Blue (PAS-AB) staining on the SGM and now include this as Supplementary Figure 7 (Fig. S7).

4. P8 paragraph 2: 'P2Y2 expression maintained consistent expression through day 14' – I find this a very strong statement, based on the variability of data in Figure 4c. Can you tone this down?

We agree that this statement was overly strong. We have now updated it to 'P2Y2 expression shows a decreasing trend at day 14 but due to significant variability, expression is not statistically different from Day 0.' See P8 paragraph 2 for highlighted changes.

5. Fig4K: can the authors make the fluorescence panels larger – hard to make out the change in colour from such small images.

Thank you for this suggestion. The size of Fig. 4k has been increased in the revised submission.

6. Fig. 5n – could the authors show the raw data of the human amylase secretion work, if possible?

Thank you for this suggestion. Unfortunately, due to the challenges of including an amylase standard curve, we are unable to provide absolute quantification of the data. However, we now report in Supplementary Figure 10 (Fig. S10) the average fluorescent intensity used to perform the normalization obtained in Fig. 5n.

REVIEWERS' COMMENTS:

Reviewer #1 (Remarks to the Author):

The authors have sufficiently addressed my concerns.

Reviewer #2 (Remarks to the Author):

Benoit and colleagues have addressed several concerns from my peer review and the manuscript is now more focused and clear from abstract to discussion. Their paper is now more novel and focused on testing a mSG organoid developed by microbubble (MB) as a proof of concept for high-content and high-throughput drug screening (see page 2, first paragraph). They added more relevant data in regards to fractionated irradiation models. Yet (as the authors repeatedly mentioned), this MB organoid technology has challenges regarding the acinar phenotype and secretory function that may or may not affect its efficacy towards its final applications. The salivary gland regeneration/bio-engineering field will have to wait for their ongoing studies to address those challenges, which are no longer within the focus of this submitted work.

Reviewer #3 (Remarks to the Author):

My comments on the suggested revision are under the replies from the authors in each relevant section. All my major revision requests have been answered, and my minor comments for the most part. Depending on the opinions of the other reviewers, I would accept this article for publication if the authors agree to make one last adjustment to the wording in the abstract (see Major Comment 2, in attached file).

COMMSBIO-20-1718-T

We would like to thank the reviewers again for their thorough and thoughtful re-reviews of our manuscript. Please find below our point-by-point responses highlighted in yellow to address additional critiques, including reference to changes made within the revised manuscript. Note that prior critiques and responses are in grey.

Reviewer #3 (Remarks to the Author):

Song et al Nature Biology Communications Review

Comments:

Song et al present an interesting manuscript addressing a problem central to salivary gland (SG) research in the context of radiation-induced damage, namely that saliva producing cells, the acinar cells, are prone to loss of phenotype in in vitro culture situations. This interferes with our ability as a radiation research community to screen potential new drug targets, and ameliorate radiation-induced hyposalivation. The manuscript is well written. I have some major and minor comments regarding the manuscript, that I would like to see addressed before publication in Nature Biology Communications.

Major:

1. Whilst I appreciate that the data of Song et al has significantly improved the culture conditions necessary to maintain SG acinar cell phenotype in vitro, the nature of the paper message, i.e. that acinar cells lose their phenotype in vitro, is not per se novel. The authors have acknowledged this themselves. I question whether this is novel enough for Nature Biology Communications.

We acknowledge that the central message of this manuscript was not clearly recognized in our initial submission. We have revised the manuscript to stress the two major advances in this work: (1) When intracellular contacts are maintained during isolation and cultured under the conditions described, the acinar cells maintain secretory phenotype longer than in previous studies; (2) arrays of these cells cultured in MBs allow us to characterize phenotypic and functional parameters in a high-throughput manner, as well as to employ them for drug testing. This level of screening has not previously been demonstrated for salivary gland tissue.

Reviewer: I am satisfied with this revision.

2. Continuing on this theme, the abstract does not in my opinion truly reflect the data presented. Namely that with all markers and experiments examined, acinar cells still lose their phenotype in vitro. The abstract suggests that acinar phenotype is maintained exactly as that of the native tissue, something I would like to be addressed.

We appreciate the reviewer's critique and have revised the abstract to state both the goal and the outcomes of this project more clearly.

Reviewer: Although the authors state that the abstract has been adjusted, it still reads as 'maintained' expression of acinar cell markers. I would still like to see this comment addressed.

We would like to thank the reviewer's comment, and have revised the sentences into 'express key salivary gland markers'.

3. There is some confusion with reference in ductal cell types in the manuscript. Namely:

i) Are the authors claiming that they do not seed striated duct-type structures, which they state are

present in their isolated cell types, into the MBs? Are they excluding these structures somehow from their analysis?

We agree with the reviewer that the approach used to isolate cells for SGm development was unclear. The cell clusters seeded into the MB arrays included all cell types that were present after the isolation and size selection were done. This most likely included striated duct structures, such as that seen in Figure S2c, which is stained with the luminal duct cell marker, K7. The only ducts which might be excluded could be the largest excretory ducts, if they are not sufficiently dissociated. However, no effort was made to purposefully exclude them. The first paragraph of the results has been revised accordingly to ensure clarity of this point.

Reviewer: I am satisfied with this revision.

ii) In the mouse system, the authors use K5 and K7 interchangeably to denote 'ductal' cells (page 6, end of second paragraph, p7 paragraph 1). Please specify precisely to which ductal cell subpopulation you refer here. K5 is commonly used for example to denote luminal cells of the striated ducts, and K7 for IDs cells and luminal striated duct cells. If the authors wish to use both markers, in the case of Fig. 2m, please perform then qPCR and staining for both K5 and K7.

We acknowledge the interchangeable use of K5 and K7 in the initial submission. We have clarified the use of both markers in the resubmission and have also included immunostaining and qPCR for both K5 and K7. Please find the added data in Fig. 2 and associated discussion in the middle of p. 7. These data show inconsistencies between K7 and K5 expression and immunostaining, whereby expression levels reach 6- and 12-fold day 0 values, yet immunostaining suggests only modest positivity within the SGm. This can be explained by cell outgrowth from the MB observed beyond day 7 of a cytokeratin-expressing population that resides on top of the chip (see Supp Fig. 8). RNA is extracted from the entire chip rather than only the SGm, therefore, the inconsistency may be due to the outgrowth cells overwhelming gene expression trends for K5 and K7.

Reviewer: I am satisfied with this revision.

iii) The authors state that K5 expression increases in time in the MB system. Again, with reference to the type of ductal cell you are referring to, what is the inference from this statement? Which type of ductal cells are increasing in prevalence? (Page 6).

It has been established that K5 and K7 (new data Fig. 2) expression increases with culture of salivary gland cells in vitro, as well as in injured or irradiated glands [Watanabe et al. 2017; Sullivan et al. 2005], and disease [Pringle et al 2020]. We have previously reported this in hydrogel-cultured spheres [Shubin et al. 2019]. It is most likely that cells activate K5 and K7 expression under stress [Pan et al. 2013 PMID: 23270662; Toivola et al. 2015 PMID: 25599598], which may result in outgrowth, as observed in Supp. Fig. 8.

Reviewer: I thank the authors for discussing my comment, but miss inclusion of this information in the manuscript, or an explanation for it not being incorporated (word count limit ,etc).

We appreciate the reviewer's suggestion and have included this information in the revised manuscript (page 6).

4. What is the authors hypothesis about the Mist1 expression? They state that Mist1 expression increases in MB culture (Day 0 24% to Day 4/6 40 %), whilst acinar cell functional markers decrease. Based on the previous work of the Ovitt group, are the authors insinuating that a acinar cell progenitor type population results from their culture system, at least partially?

We apologize that these results were confusing with regard to the overall conclusions of the paper, and have removed the quantification for these experiments from the manuscript. The results show that at both day 4 and 6 there were some cells that expressed Cre and responded to tamoxifen. This indicates that

the SGm do maintain some Mist1-positive cells. However, the values reflected the number of MBs that contained RFP positive cells on each day tested, and the data were obtained from separate isolations. They should therefore not have been directly compared and should not be interpreted as measures of Mist1 expression levels. Unfortunately, due to COVID-19 related lab shutdowns and restrictions, these experiments cannot be replicated in a reasonable timeframe. As the findings are not crucial to the overall study conclusions, we have removed this data from the resubmission.

Reviewer: I am satisfied with this revision.

Minor

1. Use of LIVE-DEAD stain: I don't agree with the method of quantification. '>90% of MBs within each array contain viable cells' – this could be anything from 1% to 100% viable cells. Is there any way you can quantify the percentage of viable cells in the MBs? Find this data a little misleading otherwise.

We agree that the LIVE/DEAD quantification is unconventional. However, due to multicellular 3D aggregates that form within the chips, robust LIVE/DEAD quantification by enumeration of individual cells is challenging and fraught with inconsistency. This is due to non-discrete cells, signal bleed through by different levels of tissue, and constraints with imaging two different channels with different integration times, whereby neither intensity level nor excitation efficiency is identical. Therefore, we elected to employ the alternative yes/no on a per MB basis, which simplifies quantification and alleviates these inconsistencies. However, it is clear from the representative images (see Fig. 1 j-k, Fig. 5a-c) that the majority of cells of the imaged tissues in individual MBs are alive.

Reviewer: This comment has not been addressed in terms of adjustment of data etc, but I can understand the technical difficulties involved and am willing to accept this.

We appreciate the reviewer's understanding and we are currently focusing on new methods to quantify many outcomes, including cell viability, with rigor and reproducibility within MB culture.

2. Fig3e-f: Amylase staining at D14 looks really bright, although authors state that it is less than Day 0. Could a Day 0 panel be added to this figure for comparison?

Thank you for suggesting this important control for comparison. The Day 0 Amylase control is now included in Fig. 3 along with Day 0 Pip controls.

Reviewer: I am satisfied with this revision.

3. Gross morphology of the acinar cells in MB chips still not extremely convincing, although markers are expressed. I miss the characteristic triangular acinar cell shape. Have you tried doing an H&E staining on the MB?

The characteristic triangular shape of acinar cells is quite prominent in AIDUCs shown in Supplemental Figures 2a, b, and d. However, as has been shown by many groups, the morphology of acinar cells is not well maintained in vitro under various culture conditions [see Srinivasan 2016 PMID: 28170182; Pringle et al. 2016 PMID: 26887347; Adine et al. 2018; Seo, Tran 2019; Nam et al. 2019 PMID: 31285850; Shubin et al. 2017 PMID: 28039063], although these reports showed maintenance of acinar cell-specific gene expression, suggesting that acinar cells are present.

We have also performed Periodic acid-Schiff's – Alcian Blue (PAS-AB) staining on the SGm and now include this as Supplementary Figure 7 (Fig. S7).

Reviewer: Apologies for the oversight, Figures 2,a,b, and d do indeed show convincing acinar cell morphology.

4. P8 paragraph 2: 'P2Y2 expression maintained consistent expression through day 14' – I find this a very strong statement, based on the variability of data in Figure 4c. Can you tone this down?

We agree that this statement was overly strong. We have now updated it to 'P2Y2 expression shows a decreasing trend at day 14 but due to significant variability, expression is not statistically different from Day 0.' See P8 paragraph 2 for highlighted changes.

Reviewer: I am satisfied with this revision.

5. Fig4K: can the authors make the fluorescence panels larger – hard to make out the change in colour from such small images.

Thank you for this suggestion. The size of Fig. 4k has been increased in the revised submission.

Reviewer: I am satisfied with this revision.

6. Fig. 5n – could the authors show the raw data of the human amylase secretion work, if possible?

Thank you for this suggestion. Unfortunately, due to the challenges of including an amylase standard curve, we are unable to provide absolute quantification of the data. However, we now report in Supplementary Figure 10 (Fig. S10) the average fluorescent intensity used to perform the normalization obtained in Fig. 5n.

Reviewer: This request has been partially addressed. I'm still not sure why producing the amylase standard curve is challenging and truly quantifying the data, or per se that Fig S10 really improved my understanding of this.

We would like to thank the reviewer's suggestion. We are adapting an amylase assay kit to work in-chip, so amylase production in each individual microbubble can be assessed. Microbubbles are seeded using gravity – therefore, we cannot directly control how much amylase falls into individual microbubbles when creating a standard curve. Positive controls with purified amylase protein and negative controls with the amylase reagent only were performed to confirm amylase positivity in SGM-containing microbubbles but the exact concentration within each microbubble is unknown, thus creation of a trustworthy standard curve is impossible.